# MOOSE-Chem: Large Language Models for Rediscovering Unseen Chemistry Scientific Hypotheses

**Zonglin Yang**[1,2*], **Wanhao Liu**[2,3], **Ben Gao**[2,4], **Tong Xie**[5,6], **Yuqiang Li**[2],
**Wanli Ouyang**[2], **Soujanya Poria**[7], **Erik Cambria**[1†], **Dongzhan Zhou**[2†]
[1] Nanyang Technological University [2] Shanghai Artificial Intelligence Laboratory
[3] University of Science and Technology of China [4] Wuhan University [5] University of New South Wales
[6] GreenDynamics [7] Singapore University of Technology and Design
{zonglin.yang,cambria}@ntu.edu.sg, zhoudongzhan@pjlab.org.cn

## Abstract

Scientific discovery plays a pivotal role in advancing human society, and recent progress in large language models (LLMs) suggests their potential to accelerate this process. However, it remains unclear whether LLMs can autonomously generate novel and valid hypotheses in chemistry. In this work, we investigate whether LLMs can discover high-quality chemistry hypotheses given only a research background—comprising a question and/or a survey—without restriction on the domain of the question. We begin with the observation that hypothesis discovery is a seemingly intractable task. To address this, we propose a formal mathematical decomposition grounded in a fundamental assumption: that most chemistry hypotheses can be composed from a research background and a set of inspirations. This decomposition leads to three practical subtasks—retrieving inspirations, composing hypotheses with inspirations, and ranking hypotheses—which together constitute a sufficient set of subtasks for the overall scientific discovery task. We further develop an agentic LLM framework, *MOOSE-Chem*, that is a direct implementation of this mathematical decomposition. To evaluate this framework, we construct a benchmark of 51 high-impact chemistry papers published and online after January 2024, each manually annotated by PhD chemists with background, inspirations, and hypothesis. The framework is able to rediscover many hypotheses with high similarity to the groundtruth, successfully capturing the core innovations—while ensuring no data contamination since it uses an LLM with knowledge cutoff date prior to 2024. Finally, based on LLM's surprisingly high accuracy on inspiration retrieval, a task with inherently out-of-distribution nature, we propose a bold assumption: that LLMs may already encode latent scientific knowledge associations not yet recognized by humans.

## 1 Introduction

Discovering new science has long been one of the deepest desires of humanity, which can not only satisfy our curiosity to understand the universe but also contribute largely to the prosperity of human society (Coccia, 2019). Recently, there are some breakthroughs indicating that LLMs have the potential to assist scientists in accelerating the discovery process (Luo et al., 2025).

Yang et al. (2024b) first find that LLMs can generate novel and valid enough hypotheses evaluated by experts. They focus on the social science domain and make discoveries by developing a multi-agent system, leveraging an assumption that a majority of social science hypotheses can be divided into a research background concept and an inspiration concept. This assumption is largely valid because a social science hypothesis is about how an independent variable can influence another dependent variable (Hair et al., 2007).

---

*Contribution during internship at Shanghai Artificial Intelligence Laboratory. †Corresponding author.

Si et al. (2024) further validate this finding by employing a large group of scientists to evaluate LLMs' generated hypotheses in the NLP domain and show that LLM can generate more novel but slightly less valid research hypotheses than human researchers. However, it is still unclear LLMs' scientific discovery ability in natural science such as the chemistry domain.

Sprueill et al. (2023; 2024) adopt LLMs to conduct a search process for catalyst discovery. However, their method is limited in the catalyst discovery domain, and their evaluation relies on whether LLMs can rediscover existing commercially used catalysts, potentially influenced by a data contamination problem. As a result, it is still unclear how good LLMs are for chemistry scientific discovery.

In this paper, we investigate this central research question: Can LLMs automatically discover novel and valid chemistry research hypotheses (even at the Nature level) given only a chemistry research background (consisting of a research question and/or a background survey), without limitation on the domain of the research question? With extensive discussions with chemistry experts, we find that the assumption used in social science, that a hypothesis can be divided into background and inspiration, can also apply to a majority of chemistry hypotheses. It is not too surprising, since cognitive science research has shown that creative ideas often result from the cohesive association of two seemingly unrelated pieces of knowledge (Koestler, 1964; Benedek et al., 2012; Lee & Chung, 2024). A main difference is that chemistry might need more than one inspiration (e.g., adding several components to compose a novel chemistry system). With this key insight, we break the seemingly impossible-to-solve central question into three smaller, more practical, and executable fundamental questions that, when summed up, should be very close to a set of sufficient conditions for the central question. Specifically, the smaller questions are (1) whether LLM can identify inspiration papers that have the potential to help with the given research question; (2) given only known knowledge (from background and inspirations), whether LLMs can infer unknown knowledge that is highly likely to be valid; and (3) whether LLM can identify good hypotheses and rank them higher.

To investigate these three questions, we build a benchmark consisting of 51 chemistry papers annotated by chemistry PhD students, breaking every paper into a background, several inspirations, and a hypothesis. The goal is to rediscover the hypothesis with only the background by using LLMs trained with data up to December 2023. The papers are all published in Nature, Science, or a similar level in 2024, and they are only made public on the internet in 2024. The benchmark is designed to be similar to the Mathematical Olympiad Competition (Trinh et al., 2024), to provide several dozens of very difficult and meaningful questions to solve. Along with the benchmark, we propose a ranking task for scientific discovery (along with evaluation criteria), which has been largely overlooked in previous works (Yang et al., 2024a; Wang et al., 2024b). Ranking is important because although AI systems can generate a large number of hypotheses in a relatively short time, verifying them one by one requires a lot of experimental costs.

Motivated by this breakup into three smaller questions, we design a multi-agent framework named MOOSE-CHEM for chemistry scientific discovery. It in general includes three stages: (1) searching through chemistry literature to find inspiration papers, (2) leveraging the inspirations to propose hypotheses for the background research question, and (3) identifying high-quality hypotheses to give them a higher rank. Compared with Yang et al. (2024b)'s method in social science that assumes a similar separation between background and inspiration for hypothesis formulation, MOOSE-CHEM adopts an evolutionary algorithm to foster a broader diversity of approaches in using inspiration for background, thereby capitalizing on the benefits derived from varied mutations. In addition, MOOSE-CHEM also adopts a multi-step design to collect more than one inspirations for chemistry discovery. Finally, it uses an efficient ranking method for better reference for scientists.

We design experiments with the benchmark to test the three fundamental questions and find that LLMs are highly capable. We also test MOOSE-CHEM with the benchmark, mimicking the setting to run it in the wild by only giving a background and a corpus of up to 3000 chemistry papers to select inspiration. Even in this challenging setting, MOOSE-CHEM can still rediscover many hypotheses with very high similarity with the ground truth ones, covering the main innovations.

Overall, the contributions of this paper are:

- We provide the first mathematical derivation on how to decompose the seemingly impossible-to-solve question $P(\text{hypothesis}|\text{research background})$ into many executable and practical smaller steps. This decomposition make $P(\text{hypothesis}|\text{research background})$ possible to be practical.

- We develop a scientific discovery framework directly based on the mathematical derivation. Different from previous works, we propose an *evolutionary algorithm-based method* to better associate background and inspiration, *multi-step inspiration retrieval and composition*, and an *efficient ranking method*. In addition, the framework can be applied to chemistry and material science, which are not covered by previous methods.

- We construct a benchmark by three chemistry PhD students, consisting of 51 chemistry papers published on Nature, Science, or a similar level, decomposing each paper into the research background, inspirations, and hypothesis.

- We propose an assumption, grounded in preliminary experiments, that LLMs may already possess numerous knowledge pairs capable of being associated to create novel knowledge—even when scientists have not previously recognized any relationship between them.

- For the first time, we show that an LLM-based framework can largely rediscover the main innovations of many chemistry hypotheses that have been published in Nature and Science. The rediscovery is not because of data contamination, because we have controlled the date of the training corpus of the LLM and the online date of the chemistry papers.

## 2 RELATED WORK

Zhong et al. (2023) work on finding the difference between two corpora to propose hypotheses, but their evaluation is conducted by Turkers, which cannot lead to a novel discovery. Wang et al. (2024b) try to utilize LLMs to discover novel NLP and biochemical hypotheses, and find the hypotheses still fall far behind scientific papers in terms of novelty, depth, and utility. Yang et al. (2024b) first show that LLMs can generate novel and valid enough hypotheses evaluated by PhD students, but they only focus on social science. FunSearch (Romera-Paredes et al., 2024) can discover specific solutions for mathematical conjecture but can't discover new math theorems. Qi et al. (2024) analyzes LLM's ability for scientific discovery in the biomedical domain by directly generating hypotheses with only the research background. Boiko et al. (2023); Baek et al. (2024); Li et al. (2024); Lu et al. (2024) focus on subsequent steps for scientific discovery, mainly developing and conducting experiments. Sprueill et al. (2023; 2024) focus on catalyst discovery, but their evaluation relies on whether can rediscover existing commercially used catalysts, which might cause data contamination problem. Kumar et al. (2024) compare different LLMs on scientific discovery in different disciplines. Tshitoyan et al. (2019) show that word embedding obtained from large-scale chemistry literature can recommend materials years before their discovery. Xie et al. (2024) predict emerging thermoelectric materials by summarizing the sentiment in the existing literature.

## 3 BENCHMARK CONSTRUCTION

The goal of the benchmark, named TOMATO-Chem, is two-fold. Firstly, it is used to analyze LLM's ability in terms of the three smaller questions. Secondly, it serves as a challenge to rediscover nature-level chemistry hypotheses with only a research background. The setting of the challenge is very similar to a real copilot setting, where scientists tell the copilot about the specific research question they are interested in, and optionally a small survey consisting of several paragraphs summarizing the existing best-performing methods for the research question.

To achieve the goals, we split each collected paper into the following components: <*background question*, *background question (strict)*, *background survey*, *background survey (strict)*, *one to three inspiration paper titles and their reason to serve as an inspiration*, *research hypothesis*, *experiments*, *reasoning process*, *summarization of inspirations*>. Every component is described by text.

The reason we add a *strict* version for *background question* and *background survey* is that many hypotheses are making relatively minor modifications based on existing methods covered by the survey, and the question can be very insightful to provide a hint on the general direction of the hypothesis. In practice, these situations are entirely possible, especially when the scientist users can provide a more comprehensive survey on existing methods, or contain deep insights in their question. Here, we also keep the strict version to make the task more challenging and encourage developing methods to better assist scientists even when they are also new to their research topic.

The *reasoning process* indicates the relation between the components of background, inspirations, and hypothesis. For example, the reasoning process can be "background + inspiration 1 + inspiration 2 = hypothesis", or "background + inspiration 1/inspiration 2 + inspiration 3 = hypothesis".

The benchmark consists of 51 chemistry and material science papers and is constructed by multiple chemistry PhD students. We only select those papers published on top chemistry venues and be public on the internet after January 2024. After constructing, the experts check again on (1) whether the identification of the inspirations is correct and whether more inspirations are needed; (2) whether the background does not contain any information in inspirations or hypothesis; and (3) whether the background and the identified inspirations can roughly logically lead to the hypothesis. The complete instruction on the check process is shown in § A.3.

| Category | Count |
|---|---|
| Polymer Chemistry | 21 |
| Organic Chemistry | 22 |
| Inorganic Chemistry | 3 |
| Analytical Chemistry | 5 |
| Total | 51 |

Table 1: Distribution of categories.

| Publication Venue | Count |
|---|---|
| Nature / Science | 27 |
| Nature Subjournals | 20 |
| Other Top Journals | 4 |
| Total | 51 |

Table 2: Distribution of publication venues.

Table 1 and Table 2 show the statistics of the benchmark in terms of chemistry category and publication venue. Material science is a sub-category of chemistry and can belong to the categories in Table 1, such as polymer material and organic material. Around 13 collected benchmark papers are inside the material science domain. Beyond them, more papers have intersections with material science. In this paper, we target both chemistry and material science, but for simplicity, we only refer to them as chemistry in this paper.

## 4 METHODOLOGY

### 4.1 FUNDAMENTAL ASSUMPTION AND FOLLOWING DECOMPOSITION

We propose an assumption that a majority of chemistry hypotheses can originate from a research background and several inspirations. This assumption is not only supported by many chemistry researchers whom we have extensive discussions with but also by the cognitive science finding that "creative ideas often result from the cohesive association of two (or more) seemingly unrelated pieces of knowledge" (Koestler, 1964; Benedek et al., 2012; Lee & Chung, 2024). We design our method based on this fundamental assumption.

Denoting background knowledge as $b$, inspiration knowledge as $i$, and hypothesis as $h$, we translate this assumption as:

$$h = f(b, i_1, \ldots, i_k) \tag{1}$$

Here, $k \in \mathbb{Z}$ represents the number of inspirations needed for a particular $h$. Typically in chemistry, $k \in [1, 3]$. In other words, given existing knowledge in the background, a majority of chemistry research is about searching knowledge that previously *not known* to be related to the background but in fact *can assist* the background, then associate the background knowledge and the searched knowledge in a reasonable way to compose a hypothesis.

Based on this assumption, we can transform the seemingly impossible-to-solve $P(h \mid b)$ into an equivalent form, where each step in the equivalent form is practical and executable.

$$P(h \mid b) \approx \prod_{j=1}^{k} P(i_j \mid b, h_{j-1}, I) \cdot P(h_j \mid b, i_j, h_{j-1}), \text{ where } h_0 = \emptyset \tag{2}$$

Here, $I$ denotes the full (chemistry) literature, representing the full inspiration space to search for every single $i$. The full proof along with detailed analyses is shown in § A.1, which is the core of this paper (and therefore highly recommend to take a read).

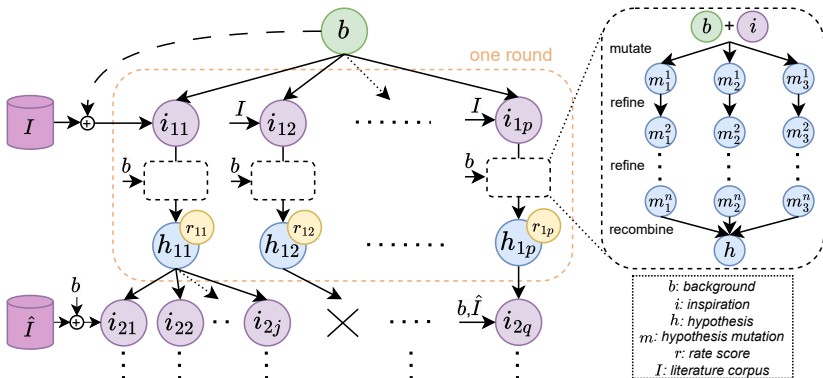

Figure 1: The MOOSE-Chem framework. It receives $b$ and $I$ as input, and outputs a list of ranked $h$. The bottom-right legend describes the symbols in the figure.

Equation 2 is meaningful in that by decomposing $P(h \mid b)$ into more practical and executable smaller questions, the seemingly impossible-to-solve $P(h \mid b)$ itself becomes practical. We analyze how $P(i_j \mid b, h_{j-1}, I)$ and $P(h_j \mid b, i_j, h_{j-1})$ are practical and executable by LLMs in § 5.1 and § 5.2 correspondingly.

Now we have clarified the steps to obtain $h$ from $b$. However, it still might not be enough helpful in practice, since $I$ can be on a large scale, and the search process might find lots of $i$, and finally lead to lots of $h$. Moreover, it is very time-consuming for scientists to conduct experiments to verify every single $h$. Therefore, it would be very helpful if the generated $h$ could be ranked based on quality. Here, we adopt a straightforward and efficient way for ranking. Specifically, we design a rating function $R(h)$, such that $R(h) \to \mathbb{R}$. Denoting the full set of generated $h$ as $H$, we can obtain

$$P(H_{\text{ranked}}) = P(H, R), \text{ where } H_{\text{ranked}} = \{h_1, h_2, \ldots, h_n \mid R(h_i) \geq R(h_{i+1}) \text{ for all } i\} \quad (3)$$

Supported by Equation 2 and Equation 3, as a result, to model $P(h \mid b)$, the only three components we need to model are $P(i_j \mid b, h_{j-1}, I)$, $P(h_j \mid b, i_j, h_{j-1})$, and $R(h)$. The implementation details of the three components are illustrated in the remaining subsections in § 4. Analyses of LLM's ability on the three components are provided in § 5.

### 4.2 THE FRAMEWORK DEVELOPED BASED ON THE ASSUMPTION

#### 4.2.1 THE GENERAL PICTURE

Our methodology is developed based on the fundamental assumption discussed in § 4.1. Specifically, we use LLMs to perform $P(i_j \mid b, h_{j-1}, I)$, $P(h_j \mid b, i_j, h_{j-1})$, and $R(h)$, and organize them into a multi-agent LLM-based framework. The input to the framework is only a *background question* and/or *background survey*, together with a (large) chemistry literature corpus to search for *inspiration*. The output of the framework is a list of ranked *research hypothesis*.

The framework's design is shown in Figure 1 (overview in Figure 2). It is a direct implementation of Equation 2 and 3. We develop it as simply as possible, retaining only the necessary parts.

In the general picture, given a research background $b$ (*research question* and/or *research survey*), the framework first performs $P(i_1 \mid b, h_0 = \emptyset, I)$ by screening through the literature corpus $I$ to select many papers $i$, where each of them has the potential to serve as an inspiration. Then the framework performs $P(h_1 \mid b, i_1, h_0 = \emptyset)$, associating $b$ and each $i$ together to compose $h$. Then, it ranks $h$ by assigning an evaluation score $r$ on each of $h_1$ by $R(h_1)$. We call these three steps as one round. Another round means going through the three steps again, based on the previous round's results.

Since normally in chemistry, no more than three inspirations are needed for one hypothesis ($k \in [1, 3]$), the default setting for MOOSE-Chem is to perform three rounds for each $b$. In every other round, the number of $i$ and $h$ can expand exponentially. Here, we adopt beam search to select a fixed size of the top-ranked $h$ to enter the next round. The default beam size is 15.

### 4.2.2 Design Details of $P(i_j \mid b, h_{j-1}, I)$ And Its Motivation

We use LLMs to conduct a screening process for $P(i_j \mid b, h_{j-1}, I)$. Specifically, for each inference, we (1) sequentially select a fixed number of papers from $I$, where the fixed number is called the screening window size (default is 15); (2) set up a prompt consisting of $b$, the title and abstract of the selected papers from $I$, and the previous $h$ (if it is not $\emptyset$); and (3) instruct the LLM to generate three titles from the input that can best serve as $i$ for $b$ (and optionally previous $h$), and give reasons.

In particular, we use LLMs to choose potential inspiration $i$, but not choose $i$ from citation nor semantic neighbors because $i$ is supposed to be previously not known to be related to $b$ (we have discussed it in § 4.1). If the chosen $i$ is already known to be related to $b$, then the composed $h$ probably would not be novel. If the chosen $i$ contains similar semantic information with $b$, then probably it is not necessary to add $i$ at all, since it does not introduce much (any) extra information.

Our bold assumption here is that advanced LLMs, trained on vast scientific literature, may already recognize novel knowledge pairs unknown to any scientist that can be associated to create novel knowledge. However, this may not be too bold, as Tshitoyan et al. (2019) showed that unsupervised word embeddings from 3.3 million materials science abstracts could predict functional materials years before their discovery. Here, the functional applications can be seen as $b$, and the recommended materials can be seen as $i$, or even directly as $h$ if it is enough similar. It probably indicates that LLMs trained with significantly more literature tokens and parameters might already be able to identify the relation between many knowledge pairs that are unknown to be related by any scientist. We analyze this assumption in § 5.1.

### 4.2.3 Design Details of $P(h_j \mid b, i_j, h_{j-1})$ And Its Motivation

The retrieved $i$ is expected to be not known to be related to $b$; therefore, it might be difficult to figure out an effective way to associate $b$ and $i$ together to compose $h$. Think of the time when backpropagation is about to be invented. Even if we are very familiar with $b$ (multi-layer logistic regression) and have successfully retrieved $i$ (chain rule in mathematics), can we invent backpropagation?

Our answer is, at least we might need to try multiple times and various ways to leverage the chain rule for multi-layer logistic regression. With this motivation, we develop a simple evolutionary algorithm-based method, shown in the top-right of Figure 1. We call it "evolutionary unit" (EU).

Specifically, given $b$ and $i$, EU will first generate multiple hypothesis "mutations" $m$, where each $m$ is a unique way to associate $b$ and $i$ together. Then EU further develops each $m$ independently by providing feedback to each $m$ in terms of validness, novelty, clarity, and significance, and then refining them based on the feedback. Yang et al. (2024b) first propose to provide feedback in terms of validness, novelty, and clarity to refine hypotheses. Here, we add an additional aspect, significance, since significance is an important evaluation criterion in chemistry. We assume the refined hypothesis should be of better quality so that the refined hypothesis is "selected", while the previous hypothesis is "eliminated" by the "environment". Finally EU "recombines" the remaining selected $m$, leveraging the advantages from every $m$ to propose $h$ to better associate $b$ and $i$.

### 4.2.4 Design Details of $R(h)$ And Its Motivation

We adopt a simple and efficient way for $R(h)$, which is to prompt an LLM to output evaluation scores for an input $h$ in terms of validness, novelty, significance, and potential. Validness and novelty are two fundamental requirements for such an inductive reasoning process as scientific discovery (Yang et al., 2024a;b). Significance is added because it is important for chemistry. We additionally add potential, because the generated $h$ are about to be further developed by scientists, so we might want to pick those $h$ that not only are currently in high quality but also have good potential to be further developed. We did not design $R(h)$ in a more complicated way, since there are lots of $h$ to rank, and we might want to save more inference time.

Yang et al. (2024b) use the scores as automatic evaluation for generated social science hypotheses and have shown a high consistency score between automatic evaluation and expert evaluation. However, in the chemistry domain, LLMs might not be reliable enough to directly evaluate the generated $h$ (Sprueill et al., 2024). But it might still be able to provide a preliminary quality identifier to $h$: the ranking of the average score between the four aspects of an $h$ determines whether it will enter the

| Corpus Size | Hit Ratio (top 20%) | Hit Ratio (top 4%) | Hit Ratio (top 0.8%) | Hit Ratio (top 0.016%) |
|---|---|---|---|---|
| 150 | 92.8% | 76.8% | 61.4% | NA |
| 300 | 96.7% | 83.7% | 60.8% | NA |
| 1000 | 96.4% | 88.9% | 69.0% | 46.7% |
| 3000 | 95.8% | 86.9% | 70.6% | 52.0% |

Table 3: Main table for $Q1$. For each screen window of 15 papers, 3 papers are selected.

| Screen window size | Hit Ratio (1 round) | Hit Ratio (2 round) | Hit Ratio (3 round) | Hit Ratio (4 round) |
|---|---|---|---|---|
| 10 | 98.0% | 88.9% | 79.4% | 56.5% |
| 15 | 96.7% | 83.7% | 60.8% | NA |
| 20 | 91.2% | 76.8% | 58.8% | NA |
| 40 | 88.9% | 54.9% | NA | NA |
| 60 | 71.6% | 53.9% | NA | NA |

Table 4: Ablation table on screen window size for $Q1$. The corpus size is 300. For each screen window no matter its size, 3 papers are selected to remain for the next round of screening.

next round of MOOSE-Chem by beam search. To understand how well LLMs can perform $R(h)$, we analyze "how well LLMs can rank chemistry hypotheses" in § 5.3.

# 5 INVESTIGATION ON FUNDAMENTAL QUESTIONS

$P(h \mid b)$ can be understood as the task to discover high-quality chemistry *research hypothesis*, given only a *background question* and/or *background survey*. Our central question to investigate is how well LLMs can perform $P(h \mid b)$. Supported by Equation 2 and 3, we break up this main question into three smaller questions: how well can LLMs perform (1) $P(i_j \mid b, h_{j-1}, I)$, (2) $P(h_j \mid b, i_j, h_{j-1})$, and (3) $R(h)$? All experiments are performed by GPT-4o (its training data is up to October 2023).

## 5.1 HOW WELL CAN LLMS PERFORM $P(i_j \mid b, h_{j-1}, I)$?

Here, we investigate the question (denoted as $Q1$): "whether LLM can identify inspiration papers which *are unknown* to be able to associate with the background (or at least unknown to associate in a certain way) but in fact can associate with the background to create novel knowledge?".

We first find 3000 most cited chemistry papers published in Nature, and construct a series of $I$ in size of 150, 300, 1000, and 3000. $I$ is constructed by first adding the ground truth inspiration papers (around 120), then randomly selecting the remaining papers from the 3000 papers, and finally randomizing the order of all the collected papers. Only title and abstract are needed for each paper in $I$. The default setting is that each inference of LLMs will screen 15 papers from $I$, and generate three titles that LLMs think can best assist $b$ (and/or previous $h$). Screening through $I$ for one round, only 20% of $I$ will be selected. Screening another round will only leave 4%, and so on.

We use Hit Ratio as the evaluation metric, which is calculated by the number of selected ground truth inspiration papers divided by the number of all ground truth inspiration papers. All the Hit Ratio numbers shown in the tables are averaged across the 51 papers in the benchmark.

Table 3 shows the main experiment results. The Hit Ratio is surprisingly high: More than 75% of the ground truth inspirations are covered by even only the 4% chosen papers from the chemistry literature corpus. It seems that LLMs are quite capable of finding inspiration papers that are unknown to be able to associate with the background but in fact, can associate with the background to create novel knowledge. It means our bold assumption in § 4.2.2 that "the most advanced LLMs might already know lots of knowledge pairs that are able to associate to create novel knowledge, where the knowledge pairs are not known by any scientist to be related" is possible to be true.

Table 4 shows the ablation study in terms of screen window size. It seems that a smaller window size can lead to better performance: a screen window size of 60 to keep 3 for one round will select 5% of the corpus, and the Hit Ratio is 71.6%; while a screen window size of 15 to keep 3 for two rounds will select only 4% of the corpus, but the Hit Ratio is as high as 83.7%.

| Model | Hit Ratio (top 20%) | Hit Ratio (top 4%) | Hit Ratio (top 0.8%) |
|---|---|---|---|
| `Llama-3.1-8B` | 71.6% | 43.5% | 26.8% |
| `Llama-3.1-70B` | 95.1% | 83.0% | 59.5% |
| `Llama-3.1-405B` | 95.7% | 78.7% | 52.7% |
| `GPT-4o` | 96.7% | 83.7% | 60.8% |

Table 5: Comparison of `Llama` series and `GPT-4o` on inspiration retrieval. The corpus size is 300. For each screen window of 15 papers, 3 papers are selected.

| 5 points | Generated hypothesis covers three key points (or covers all the key points) and leverage them similarly as in the groundtruth hypothesis; Extra key points do not have apparent flaws. |
|---|---|
| 4 points | Generated hypothesis covers three key points (or covers all the key points) and leverage them similarly as in the groundtruth hypothesis; Extra key points have apparent flaws. |
| 3 points | Generated hypothesis covers two key points and leverage them similarly as in the groundtruth hypothesis, but does not cover more or all key points |
| 2 points | Generated hypothesis covers one key point and leverage it similarly as in the groundtruth hypothesis, but does not cover more or all key points |
| 1 point | Generated hypothesis covers at least one key point, but is used differently as in the groundtruth hypothesis |
| 0 point | Generated hypothesis does not cover any key point |

Table 6: Description of the Matched Score.

Table 5 compares LLMs in different scales on inspiration retrieval ability. The results indicate that LLMs obtain the emergent ability for inspiration retrieval since a rather small parameter size, but then quickly plateau. § A.9 discusses research background options' influence on inspiration retrieval.

## 5.2 How Well can LLMs perform $P(h_j \mid b, i_j, h_{j-1})$?

Here, we investigate the question (denoted as $Q2$): "Given only known knowledge, whether LLM can reason to unknown knowledge that has high probability to be valid?".

The first challenge to answer $Q2$ is the evaluation method: The benchmark covers a large range of chemistry topics, and chemistry is a very complex discipline that a slight change of research topic would make a chemist unable to provide a reliable enough evaluation. In fact, a chemistry researcher might not be able to provide a reliable enough evaluation even if the hypothesis is in his domain.

Therefore, we adopt a reference-based evaluation method called "Matched Score" (MS). The descriptions are shown in Table 6. It's on a 6-point Likert scale, roughly containing four stages. Denoting generated hypothesis as $gh$, and original hypothesis as $oh$, the four stages are (1) $gh \cap oh = \emptyset$ (0 point); (2) $gh \cap oh \neq \emptyset$ (1/2/3 points); (3) $gh \supseteq oh$ (4 points); (4) $gh \approx oh$ (5 points).

We use MOOSE-Chem to investigate $Q2$. Specifically, we initialize $I$ as only the ground truth inspiration papers and search $i$ for $k$ round, where $k$ is the number of ground truth $i$ needed for each

| | 5 | 4 | 3 | 2 | 1 | 0 | Total |
|---|---|---|---|---|---|---|---|
| | w/ background survey | | | | | | |
| Average MS (`GPT-4o`) | 2 | 9 | 18 | 17 | 5 | 0 | 51 |
| Top MS (`GPT-4o`) | 28 | 1 | 19 | 3 | 0 | 0 | 51 |
| Top MS (Experts) | 9 | 12 | 22 | 6 | 2 | 0 | 51 |
| | w/o background survey | | | | | | |
| Average MS (`GPT-4o`) | 1 | 7 | 17 | 19 | 7 | 0 | 51 |
| Top MS (`GPT-4o`) | 25 | 2 | 19 | 5 | 0 | 0 | 51 |

Table 7: Main table for $Q2$. Average/Top MS means the average/highest Matched Score of all generated $h$ from one $b$. Table 12 is a more complete version of this table including automatic evaluation results by `Claude-3.5-Sonnet` and `Gemini-1.5-Pro`.

| #Matched $i$ | 3 | 2 | 1 | 0 |
|---|---|---|---|---|
| Average Rank Ratio | NA | 0.411 | 0.474 | 0.521 |
| Size | 0 | 302 | 2458 | 4899 |

Table 8: Relation between the number of matched ground truth $i$ and the average ranking ratio ($\downarrow$).

| Matched Score | 5 | 4 | 3 | 2 | 1 | 0 | -1 |
|---|---|---|---|---|---|---|---|
| Average Rank Ratio | 0.489 | 0.439 | 0.488 | 0.501 | 0.436 | 0.501 | 0.503 |
| Size | 210 | 36 | 404 | 427 | 29 | 102 | 6451 |

Table 9: Relation between the `GPT-4o` labeled Matched Score and average ranking ratio ($\downarrow$).

$b$. MOOSE-Chem will not retrieve the same $i$ already retrieved in previous rounds, guaranteeing that before generating the final $h$, the framework has already seen all the ground truth inspirations.

Table 7 shows the results. For each $b$, the top two $h$ with the highest MS by GPT-4o are selected for expert evaluation (by two chemistry PhD students). It indicates that LLMs are quite capable of associating known knowledge into unknown knowledge that has a high probability to be valid (very close to $oh$). In addition, providing a survey can assist the new knowledge-discovery process. We discuss the agreement between GPT-4o-based evaluation and expert evaluation in § A.14.

### 5.3 How Well can LLMs perform $R(h)$?

Here, we investigate $Q3$: "whether LLMs can select high-quality $h$ to rank them higher?".

To investigate $Q3$, we run MOOSE-Chem with every $b$ from the benchmark; $|I| = 300$, containing all the ground truth $i$. Every $h$ is given a rating $r = R(h)$, and is ranked based on $r$. For every generated $h$, we get the number of ground truth $i$ it leveraged (#Matched $i$), and evaluate it with a `GPT-4o` evaluated MS (here MS is -1 means this $h$ has not used any ground truth $i$).

Table 8 shows the relation between the #Matched $i$ and average ranking ratio (the lower, the better). It shows a clear trend that the more ground truth $i$ is leveraged, the better ranking score $h$ can have. It indicates that $h$ with a higher ranking ratio is more likely to be matched with better $i$.

Table 9 shows the relation between the `GPT-4o` evaluated MS and the average ranking ratio. There is a trend that the higher the MS, the better the average rank ratio (when MS $\in$ [2,4]). However, the disadvantage of those $h$ without a positive MS is not very significant. It seems that LLMs have a certain ability to rank good $h$ higher. But it is not sure how significant it is, because a part of the reason for these results is that those $h$ generated without ground truth $i$ could be also in high quality.

## 6 Experiment and Ablation Study

We perform experiments in a setting similar to the copilot in the wild setting. Only *background question (strict)*, *background survey (strict)*, and a chemistry corpus $|I| = 300$ are provided to the framework. Only the top 4% of $I$ is selected and used to develop $h$. The evaluation metrics are Top MS and Average MS (the highest/average Matched Score of all generated $h$ from one $b$), averaging across the benchmark. Experiments are conducted by `GPT-4o` (training data up to October 2023).

### 6.1 Baselines

**MOOSE** is a hypothesis discovery framework for the general social science domain. It leverages LLMs to retrieve inspirations and uses self-refine (Madaan et al., 2023) to improve the validness, novelty, and clarity aspects. The difference is that (1) it does not adopt the mutation and recombination step to better associate background and inspiration; (2) it only retrieves one step of inspiration.

**SciMON** is a hypothesis discovery framework for the NLP and biochemical domain. It relies on semantic and citation neighbors to retrieve information to assist the background. As a result, the retrieved information could be very related to the background that might not be able to serve as an inspiration. To make the generated hypothesis more novel, it adopts self-refine to focus on improving

| Method | Top MS | Average MS |
|---|---|---|
| SciMON (Wang et al., 2024b) | 2.549 | 2.281 |
| MOOSE (Yang et al., 2024a) | 2.882 | 2.464 |
| Qi et al. (2024) | 2.686 | 2.356 |
| MOOSE-Chem | **4.020** | 2.564 |
| w/o multi-step | 3.765 | 2.730 |
| w/o multi-step & EU | 2.863 | 2.578 |

Table 10: Experiments and ablation study. The Matched Score (MS) is evaluated by `GPT-4o` (this table), `Claude-3.5-Sonnet` (Table 13), and `Gemini-1.5-Pro` (Table 14).

| | 5 | 4 | 3 | 2 | 1 | 0 | Total |
|---|---|---|---|---|---|---|---|
| Top MS (Expert) | 0 | 2 | 19 | 16 | 8 | 6 | 51 |

Table 11: MOOSE-Chem runs with $|I|$=300, mimicking the copilot setting. This table shows the statistics of the top Matched Score across the benchmark. The evaluation is done by experts.

the novelty aspect of the generated hypothesis. Here, we implement SciMON with LLM-based inspiration retrieval, the same as MOOSE-Chem. Table 3 shows that the recall rate of LLM-based retrieval is 83.7%.

**Qi et al. (2024)** work on hypothesis discovery in the biomedical domain. It retrieves information pertinent to the keywords in the background to generate hypotheses. As a result, the retrieved information might compose of a *background survey*, but not as inspiration. Self-refine is also adopted.

## 6.2 RESULTS

Table 10 shows the baseline results and the ablation study of MOOSE-Chem. It indicates that both mutation & recombination and the multi-step designs can significantly improve the best-performing $h$. Mutation & recombination leads to a drop of Average MS compared to the MOOSE baseline; we attribute the reason to that the mutation step forces LLMs to generate $h$ different from previous $h$ mutations from the same $b$ and $i$, and therefore might generate many $h$ that do not make a lot of sense. The assigned MS to these mutation $h$ is low, and therefore lower down the Average MS.

To better understand the performance of MOOSE-Chem in this real copilot setting, for each $b$ the top 4 generated $h$ with the highest MS by `GPT-4o` are evaluated again by two experts in terms of MS. Table 11 shows the expert evaluation results. Here, the top MS is the highest MS for each $b$, out of the 4 expert evaluated $h$ for this $b$. Note that MS rated as three is already very high. Illustrated in Table 6, it means the generated $h$ by MOOSE-Chem (that has not seen $h$) in the real copilot setting covers two main innovations of the chemistry hypothesis, which is published in Nature, Science or a similar level. Some case studies can be seen in § A.16.

## 7 CONCLUSION

We investigated this central question: "Can LLMs automatically discover novel and valid chemistry (including material science) research hypotheses (even those which deserve a publication in Nature, Science, or a similar level) given only a chemistry research background (consisting of a research question and/or a background survey), without limitation on the domain of the research question?". We proposed a fundamental assumption to break up this seemingly impossible-to-solve central question into three smaller, more practical, and executable fundamental questions. Then, we investigated LLM's ability on each of them.

To this end, we constructed a benchmark consisting of chemistry and material science papers published and only be public in 2024. We also developed an LLM-based multi-agent framework consisting of three stages reflecting the three smaller fundamental questions. Experiments showed that the framework (runs in a copilot in-the-wild setting, with LLMs with training data up to October 2023) can rediscover many hypotheses with very high similarity with the ground-truth ones, covering the main innovations.

ACKNOWLEDGMENTS

This work is supported by the Shanghai Municipal Science and Technology Major Project. This work is supported by Shanghai Artificial Intelligence Laboratory. This research/project is supported by the Ministry of Education, Singapore under its MOE Academic Research Fund Tier 2 (STEM RIE2025 Award MOE-T2EP20123-0005).

We thank Mengsong Wu for his insightful discussions with us, and we thank Yuwei Wan for her efforts to support this research.

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

# A APPENDIX

## A.1 PROOF OF THE RIGOROUS DECOMPOSITION OF $P(h \mid b)$ BASED ON THE FUNDAMENTAL ASSUMPTION

We propose an assumption that a majority of chemistry hypotheses can originate from a research background and several inspirations. This assumption is not only supported by many chemistry researchers whom we have extensive discussions with but also by the cognitive science finding that "*creative ideas often result from the cohesive association of two (or more) seemingly unrelated pieces of knowledge*" (Koestler, 1964; Benedek et al., 2012; Lee & Chung, 2024). We design our method based on this fundamental assumption.

This assumption is reminiscent of Swanson Linking (Swanson, 1986) in the domain of literature-based discovery (LBD), also known as the "ABC model", where two concepts A and C are hypothesized as linked if they both co-occur with some intermediate concept B in papers. Our assumption differs in that: (1) for a chemistry hypothesis published in a good venue, usually more than one inspirations are needed; (2) background and inspiration are not necessarily linked by a path of intermediate papers; (3) our assumption is applied to a majority of existing published chemistry hypotheses, while LBD has been considered to only focus on a very specific, narrow type of hypothesis (Wang et al., 2024b). It might indicate that a similar proportion of future chemistry hypotheses can also be resulted from linkages of existing literature.

Denoting background knowledge as $b$, inspiration knowledge as $i$, and hypothesis as $h$, we translate this assumption as:

$$h = f(b, i_1, \ldots, i_k) \tag{4}$$

Here, $k \in \mathbb{Z}$ represents the number of inspirations needed for a particular $h$. Typically in chemistry, $k \in [1, 3]$.

Equation 4 expresses the idea that, for the majority of chemistry hypotheses (if not all), each hypothesis can be formulated as a composition of background knowledge and additional knowledge elements, which we refer to as inspirations. This functional form reflects a universal pattern in hypothesis formulation: regardless of where the inspiration originates—be it prior literature, serendipitous observation, or discussions with peers—the essential step is to identify correct inspirations and integrate them with existing background knowledge in a meaningful way. In this formulation, the process of first collecting and selecting the appropriate background knowledge and then identifying and integrating suitable inspirations constitutes both a *necessary* and *sufficient* condition for generating a valid hypothesis $h$.

Here's an example in chemistry:

- Research Question: How to obtain $D_2$ gas more efficiently?
- Background Knowledge: The best performing methods are **electrocatalytic** methods.
- Inspiration Knowledge 1: **Ruthenium** as catalyst
- Inspiration Knowledge 2: **Nitrogen-doped electrode**
- Inspiration Knowledge 3: $D_2O$ as chemical solution
- Hypothesis: A **nitrogen-doped ruthenium (Ru) electrode** can effectively catalyze the reductive deuteration of (hetero)arenes in the presence of $D_2O$ in an **electrocatalytic** method, leading to efficient $D_2$ gas production.

Here's an example in AI:

- Research Question: How can we automatically update the parameters of a multi-layer logistic regression model using data?
- Background Knowledge: Multi-layer logistic regression
- Inspiration Knowledge: The chain rule from calculus

- Hypothesis: Backpropagation

Here's another example in AI:

- Research Question: How can we improve reasoning performance in language models?
- Background Knowledge: Chain-of-Thought prompting
- Inspiration Knowledge: Majority voting over multiple reasoning paths
- Hypothesis: Self-consistency decoding

Here, "background knowledge" and "inspiration knowledge" as illustrated in the example above, can be understood as specific, well-defined knowledge pieces. In practice, however, knowledge retrieval is rarely so clean. Instead of isolating a single knowledge unit, we often retrieve a noisy cluster of information that contains the desired piece along with extraneous content. For instance, when retrieving a paper that includes a relevant inspiration, the paper will inevitably also contain unrelated information that may not be useful for the current research question. Conversely, a single clean inspiration $i$ may be embedded across multiple papers in the literature. This redundancy is beneficial—it increases the likelihood of retrieving $i$ even when searching imperfectly.

In other words, given existing knowledge in the background, a majority of chemistry research is about searching knowledge that previously *not known* to be related to the background but in fact *can assist* the background, then associate the background knowledge and the searched knowledge in a reasonable way to compose a hypothesis. Crucially, the inspiration should not be previously known to be related to the background—at least not in a way that has already been used to formulate hypotheses. Otherwise, the resulting hypothesis would lack novelty. This requirement positions the inspiration retrieval task as an inherently out-of-distribution (OOD) problem, where the goal is to surface connections that lie outside established knowledge associations.

Our goal is to transform the seemingly impossible-to-solve $P(h \mid b)$ into an equivalent form, where each step in the equivalent form is practical and executable. Denoting the full inspiration knowledge space as $I$, such that $P(I) = 1$. Then a straightforward way of decomposing $P(h \mid b)$ is by the chain rule based on Equation 4:

$$P(h \mid b) = P(h, i_1, \ldots, i_k \mid b) \tag{5}$$

$$= \begin{cases} \frac{P(h,b,i_1)}{P(b,i_1)} \cdot \frac{P(b,i_1) \cdot P(I)}{P(b) \cdot P(I)} & \text{if } k = 1 \\ \frac{P(h,b,i_1,\ldots,i_k)}{P(b,i_1,\ldots,i_k)} \cdot \frac{P(b,i_1,\ldots,i_k) \cdot P(I)}{P(b,i_1,\ldots,i_{k-1}) \cdot P(I)} \cdot \ldots \cdot \frac{P(b,i_1) \cdot P(I)}{P(b) \cdot P(I)} & \text{if } k > 1 \end{cases} \tag{6}$$

$$= \begin{cases} P(h \mid b, i_1) \cdot P(i_1 \mid b, I) & \text{if } k = 1 \\ P(h \mid b, i_1, \ldots, i_k) \cdot \prod_{j=2}^{k} P(i_j \mid b, i_1, \ldots, i_{j-1}, I) \cdot P(i_1 \mid b, I) & \text{if } k > 1 \end{cases} \tag{7}$$

Here, $P(h, i_1, \ldots, i_k \mid b)$ can be interpreted as the joint distribution over $h$ and its associated inspirations conditioned on background knowledge $b$. Equation 5 holds because Equation 4 asserts that $\{i_1, \ldots, i_k\}$ constitutes a necessary and sufficient set of additional knowledge required to compose $h$. $I$ denotes the full inspiration space—that is, the set of all possible knowledge pieces that could serve as an inspiration for generating a new hypothesis. This space includes not only all existing chemistry knowledge but also potentially relevant knowledge from other disciplines that could be leveraged in formulating novel chemistry hypotheses. However, computing over the full space $I$ is computationally infeasible. To make the problem tractable, we approximate $I$ with a large but manageable subset $\hat{I}$, consisting of approximately 3,000 top cited chemistry papers from the existing chemistry literature.

Equation 7 describes the process of $P(h \mid b)$ from a knowledge-searching perspective. However, the terms $P(h \mid b, i_1, \ldots, i_k)$ and $P(i_j \mid b, i_1, \ldots, i_{j-1}, I)$ may not fully capture how chemistry researchers actually discover new inspirations in practice. One key reason is that researchers typically reason in an incremental fashion, composing hypotheses by integrating one or two knowledge components at a time. It is cognitively and practically difficult to evaluate or integrate all candidate inspirations simultaneously. Instead, researchers iteratively assess partial combinations—gradually building toward a complete hypothesis.

To mimic how chemistry researchers conduct research and make it more practicable, we break $P(h \mid b, i_1, \ldots, i_k)$ into a series of recursive smaller steps as

$$P(h_k \mid b, i_1, \ldots, i_k) \approx P(h_k \mid b, f(b, i_1, \ldots, i_{k-1}), i_k) \qquad \text{if } k > 1 \qquad (8)$$
$$= P(h_k \mid b, h_{k-1}, i_k) \qquad \text{if } k > 1 \qquad (9)$$

Similarly, we can break $P(i_{j+1} \mid b, i_1, \ldots, i_j, I)$ as

$$P(i_{k+1} \mid b, i_1, \ldots, i_k, I) \approx P(i_{k+1} \mid b, f(b, i_1, \ldots, i_k), I) \qquad \text{if } k > 1 \qquad (10)$$
$$= P(i_{k+1} \mid b, h_k, I) \qquad \text{if } k > 1 \qquad (11)$$

As a result, to achieve the final $h_k$, we need to obtain $\{h_1, \ldots, h_{k-1}\}$ first (if $k > 1$). In addition, seeing $h$ as a "state", and $i$ as an "action", obtaining $h$ and $i$ through $P(h_k \mid b, h_{k-1}, i_k)$ and $P(i_{k+1} \mid b, h_k, I)$ correspondingly indicates a Markov property: (1) a new $h$ only depends on $b$, its previous $h$, and the current $i$; and (2) an $i$ only depends on $b$, $I$, and the previous $h$.

For brevity, we visualize this as:

$$b \xrightarrow{i_1} h_1 \xrightarrow{i_2} h_2 \xrightarrow{\cdots} h_{k-1} \xrightarrow{i_k} h_k = h,$$

where each transition is still conditioned on the background knowledge $b$, though $b$ is omitted from the notation to emphasize the Markov structure of the progression.

Building on this visualization, we formalize the formation of a hypothesis $h$ (specifically, $h = h_k$) as a constructive process that sequentially integrates a set of inspirations $\{i_1, \ldots, i_k\}$ into intermediate hypothesis states $\{h_1, \ldots, h_k\}$.

Formally, the conditional probability $P(h \mid b)$ can be expressed as a marginal over all valid sequences of inspirations that can generate $h$:

$$P(h \mid b) = \sum_{\pi \in \Pi_k} P\big(i_{\pi(1)}, \ldots, i_{\pi(k)}, h_1, \ldots, h_k \mid b\big), \qquad (12)$$

where $\Pi_k$ denotes the set of all permutations of $\{1, \ldots, k\}$ applied to the inspirations $\{i_1, \ldots, i_k\}$ such that the resulting composition yields the final hypothesis $h_k = h$, under the assumption in Equation 4, which specifies that $h$ is fully determined by $b$ and $\{i_1, \ldots, i_k\}$. In this context, $h_j$ represents the intermediate hypothesis state obtained after integrating $i_{\pi(j)}$ at step $j$.

The degree to which the order of inspirations $\{i_1, \ldots, i_k\}$ affects hypothesis formulation can vary across disciplines. In empirical sciences such as chemistry, the contributions of individual inspirations are largely interchangeable, and their order of integration has limited impact on the final hypothesis, resulting in a large $|\Pi_k|$. Conversely, in disciplines like mathematics, where hypotheses (e.g., theorems) often require constructing a specific sequence of lemmas and prior results, the ordering of inspirations is more constrained and may follow a near-deterministic path.

Considering this variance depending on the disciplines, to make this problem tractable and easier to understand, we adopt a fixed constructive path assumption, i.e., $|\Pi_k| = 1$, selecting a representative order of inspirations $\{i_1, \ldots, i_k\}$.

Under this inspiration-fixed-order assumption, where hypothesis $h$ is constructed through a specific sequence of inspirations $\{i_1, \ldots, i_k\}$, we have:

$$P(h \mid b) = P(i_1, \ldots, i_k, h_1, \ldots, h_k \mid b), \qquad (13)$$

with $h_j$ denoting the intermediate hypothesis state after incorporating $i_j$, and $h_k = h$.

Therefore, if with this inspiration-fixed-order assumption for simplicity, if $k > 1$,

$$P(h \mid b) = P(i_1, \ldots, i_k, h_1, \ldots, h_k \mid b) \tag{14}$$

$$= P(i_1, h_1 \mid b) \cdot P(i_2, h_2 \mid b, i_1, h_1) \cdot \ldots \cdot P(i_k, h_k \mid b, i_1, \ldots, i_{k-1}, h_1, \ldots, h_{k-1}) \tag{15}$$

$$\approx P(i_1, h_1 \mid b) \cdot P(i_2, h_2 \mid b, h_1) \cdot \ldots \cdot P(i_k, h_k \mid b, h_{k-1}) \tag{16}$$

$$= \frac{P(b, i_1, I)}{P(b, I)} \cdot \frac{P(b, i_1, h_1)}{P(b, i_1)} \cdot \ldots \cdot \frac{P(b, i_k, h_{k-1}, I)}{P(b, h_{k-1}, I)} \cdot \frac{P(b, i_k, h_{k-1}, h_k)}{P(b, i_k, h_{k-1})} \tag{17}$$

$$= P(i_1 \mid b, I) \cdot P(h_1 \mid b, i_1) \cdot \prod_{j=1}^{k-1} P(i_{j+1} \mid b, h_j, I) \cdot P(h_{j+1} \mid b, i_{j+1}, h_j) \tag{18}$$

$$= \prod_{j=1}^{k} P(i_j \mid b, h_{j-1}, I) \cdot P(h_j \mid b, i_j, h_{j-1}), \text{ where } h_0 = \emptyset \tag{19}$$

Although starting from $k > 1$, Derivation 19 covers the situation when $k = 1$ in Equation 7.

Therefore, in sum, we break up the seemingly impossible question $P(h \mid b)$ into many practical and executable smaller questions as:

$$P(h \mid b) \approx \prod_{j=1}^{k} P(i_j \mid b, h_{j-1}, I) \cdot P(h_j \mid b, i_j, h_{j-1}), \text{ where } h_0 = \emptyset \text{ and } k >= 1 \tag{20}$$

Of course, without the inspiration-fixed-order assumption, a more complete derivation of $P(h \mid b)$ involves marginalizing over all valid permutations in $\Pi_k$:

$$P(h \mid b) = \sum_{\pi \in \Pi_k} P(i_{\pi(1)}, \ldots, i_{\pi(k)}, h_1, \ldots, h_k \mid b) \tag{21}$$

$$\approx \sum_{\pi \in \Pi_k} \prod_{j=1}^{k} P(i_{\pi(j)} \mid b, h_{j-1}^{(\pi)}, I) \cdot P(h_j^{(\pi)} \mid b, i_{\pi(j)}, h_{j-1}^{(\pi)}), \tag{22}$$

where $h_0^{(\pi)} = \emptyset$, and $k >= 1$. Here, $h_j^{(\pi)}$ denotes the intermediate hypothesis state at step $j$ in the permutation $\pi$, which results from sequentially incorporating inspirations in the order $\{i_{\pi(1)}, \ldots, i_{\pi(j)}\}$.

## A.2 MOOSE-CHEM OVERVIEW I/O FIGURE

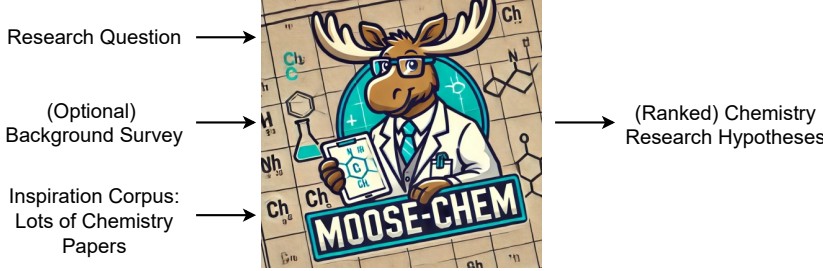

Figure 2: Overview of the input and output of the MOOSE-Chem framework.

Figure 2 shows the input and output overview of the MOOSE-Chem framework.

### A.3 The Full Instruction for Benchmark Checking

Please help us check again before finalizing the decomposition of each paper in the benchmark:

1. Whether the background question is correct.

2. Background survey shouldn't contain any information/method in inspiration or hypothesis (except if this information/method has been used for this particular background question before). It is encouraged to include the most similar existing method to the proposed method. For example, the proposal is to change BaCl2 to BaSO4. It is encouraged to include BaCl2 in the survey, but SO4 must not be included in the survey (since SO4 belongs to the inspiration).

3. Background question cannot contain any information in inspiration or hypothesis as well: It should be a little bit general question, instead of a specific question asking about how the inspiration can be leveraged to help with the question. It also shouldn't be too general that we can't understand which specific research domain it works on.

3. Whether the identification of inspirations really the main inspirations for this paper, and whether we need more main inspiration(s).

4. Whether the main hypothesis is correct and covers the main key points.

5. Whether the background survey + background question + identified inspirations can logically lead to the hypothesis (if not, we might need to identify more inspirations).

Thank you for the efforts! Your contribution is indispensable for the success of this research. Please let me know if you have any questions.

### A.4 Prompt to obtain $R(h)$

You are known as a diligent and harsh reviewer in Chemistry and Material Science that will spend much time to find flaws when reviewing and therefore usually gives a relatively much lower score than other reviewers. But when you meet with a hypothesis you truly appreciate, you don't mind to give it good scores. Given a not yet peer reviewed research hypothesis in Chemistry or Material Science domain, try to evaluate the research hypothesis from four research aspects and give score according to evaluation guidelines provided below. All four aspects should be evaluated in a 5 point scale.

Aspect 1: Validness.
5 points: The hypothesis is a logical next step from current research, strongly supported by theory, perhaps with some indirect experimental evidence or highly predictive computational results. The experimental verification seems straightforward with a high probability of confirming the hypothesis; 4 points: Here, the hypothesis is well-rooted in existing theory with some preliminary data or computational models supporting it. It extends known science into new but logically consistent areas, where experiments are feasible with current technology, and there's a reasonable expectation of positive results; 3 points: This hypothesis is within the realm of theoretical possibility but stretches the boundaries of what's known. It might combine existing knowledge in very novel ways or predict outcomes for which there's no direct evidence yet. There's a conceptual framework for testing, but success is uncertain; 2 points: While the hypothesis might be grounded in some theoretical aspects, it significantly deviates from current understanding or requires conditions or materials that are currently impossible or highly improbable to achieve or synthesize; 1 point: The hypothesis proposes concepts or outcomes that are not only unsupported by current theory but also contradict well-established principles or data. There's no clear path to experimental testing due to fundamental theoretical or practical barriers.

Aspect 2: Novelty.
5 points: This level of novelty could fundamentally alter our understanding of chemistry or create entirely new fields. It often involves predictions or discoveries that, if proven, would require a significant overhaul of existing chemical theories; 4 points: The hypothesis significantly departs from established norms, potentially redefining how certain chemical phenomena are understood or applied. It might involve entirely new materials or theoretical frameworks; 3 points: This level

involves a hypothesis that could potentially lead to new insights or applications. It might challenge minor aspects of current theories or introduce new methodologies or materials; 2 points: The hypothesis introduces a new angle or method within an established framework. It might involve known compounds or reactions but in contexts or combinations not previously explored; 1 point: The hypothesis involves minor tweaks or applications of well-known principles or techniques. It might slightly extend existing knowledge but doesn't introduce fundamentally new concepts.

Aspect 3: Significance.
5 points: This hypothesis could fundamentally change one or more branches of chemistry. It might introduce entirely new principles, theories, or methodologies that redefine the boundaries of chemical science; 4 points: This hypothesis challenges current understanding or introduces a concept that could lead to substantial changes in how a particular area of chemistry is viewed or applied. It might lead to new technologies or significant theoretical advancements; 3 points: this hypothesis proposes something new or an innovative approach that could lead to noticeable advancements in a specific area of chemistry. It might open new avenues for research or application but doesn't revolutionize the field; 2 points: This hypothesis might offer a small variation or incremental improvement on existing knowledge. It could potentially refine a known concept but doesn't significantly alter the field; 1 point: The hypothesis addresses a very narrow or already well-established aspect of chemistry. It might confirm what is already known without adding much new insight.

Aspect 4: Potential.
5 points: The hypothesis, while potentially intriguing now, holds the promise of being revolutionary with the addition of a key methodological component. This could introduce entirely new concepts or fields, fundamentally changing our understanding or capabilities in chemistry; 4 points: The hypothesis, though promising, could be transformative with the right methodological enhancement. This enhancement might lead to groundbreaking discoveries or applications, significantly advancing the field; 3 points: The hypothesis, while interesting in its current form, could be significantly elevated with the right methodological addition. This might lead to new insights or applications that go beyond the initial scope; 2 points: The hypothesis currently offers some value but has the potential for more substantial contributions if enhanced with a new methodological approach. This could lead to incremental advancements in understanding or application; 1 point: The hypothesis, as it stands, might be straightforward or well-trodden. Even with methodological enhancements, it's unlikely to significantly expand current knowledge or applications beyond minor improvements.

The hypothesis is:
Please give a response to the initial question on scoring the hypothesis from four aspects. Remember that you are a diligent and harsh reviewer.

## A.5 AUTOMATIC EVALUATION BY CLAUDE AND GEMINI

To investigate whether the results and corresponding conclusions in the main text are caused by the usage of `GPT-4o` for automatic evaluation, here we use `Claude-3.5-Sonnet` and `Gemini-1.5-Pro` to evaluate all of the results that have been evaluated by `GPT-4o`.

Table 12 covers the contents in Table 7, but with more results on using `Claude-3.5-Sonnet` and `Gemini-1.5-Pro` for automatic evaluation. When using different LLMs for automatic evaluation, the instruction is the same (can be found in § A.12). The robust results indicate again that LLMs are quite capable of associating known knowledge into unknown knowledge that has a high probability to be valid (very close to *oh*).

Table 13 and Table 14 evaluate the same hypotheses with Table 10, but using `Claude-3.5-Sonnet` and `Gemini-1.5-Pro` for automatic evaluation correspondingly (instead of `GPT-4o`). The results indicate the robustness of MOOSE-Chem and its components.

| | 5 | 4 | 3 | 2 | 1 | 0 | Total |
|---|---|---|---|---|---|---|---|
| | w/ background survey | | | | | | |
| Average MS (`GPT-4o`) | 2 | 9 | 18 | 17 | 5 | 0 | 51 |
| Average MS (`Claude-3.5-Sonnet`) | 4 | 19 | 15 | 10 | 3 | 0 | 51 |
| Average MS (`Gemini-1.5-Pro`) | 2 | 13 | 17 | 8 | 11 | 0 | 51 |
| Top MS (`GPT-4o`) | 28 | 1 | 19 | 3 | 0 | 0 | 51 |
| Top MS (`Claude-3.5-Sonnet`) | 33 | 7 | 10 | 1 | 0 | 0 | 51 |
| Top MS (`Gemini-1.5-Pro`) | 20 | 18 | 0 | 12 | 1 | 0 | 51 |
| Top MS (Experts) | 9 | 12 | 22 | 6 | 2 | 0 | 51 |
| | w/o background survey | | | | | | |
| Average MS (`GPT-4o`) | 1 | 7 | 17 | 19 | 7 | 0 | 51 |
| Average MS (`Claude-3.5-Sonnet`) | 7 | 24 | 18 | 2 | 0 | 0 | 51 |
| Average MS (`Gemini-1.5-Pro`) | 4 | 9 | 14 | 15 | 5 | 4 | 51 |
| Top MS (`GPT-4o`) | 25 | 2 | 19 | 5 | 0 | 0 | 51 |
| Top MS (`Claude-3.5-Sonnet`) | 31 | 19 | 1 | 0 | 0 | 0 | 51 |
| Top MS (`Gemini-1.5-Pro`) | 19 | 19 | 1 | 11 | 0 | 1 | 51 |

Table 12: Main table for $Q2$. Average/Top MS means the average/highest Matched Score of all generated $h$ from one $b$. The numbers represent the statistics of Average/Top MS over the benchmark.

| Method | Top MS | Average MS |
|---|---|---|
| SciMON (Wang et al., 2024b) | 3.824 | 3.529 |
| MOOSE (Yang et al., 2024a) | 3.902 | 3.559 |
| Qi et al. (2024) | 3.431 | 3.092 |
| MOOSE-Chem | **4.471** | 3.697 |
| w/o multi-step | 4.216 | 3.592 |
| w/o multi-step & EU | 3.941 | 3.614 |

Table 13: Experiments and ablation study. The Matched Score is evaluated by `Claude-3.5-Sonnet`.

| Method | Top MS | Average MS |
|---|---|---|
| SciMON (Wang et al., 2024b) | 2.980 | 2.618 |
| MOOSE (Yang et al., 2024a) | 3.039 | 2.690 |
| Qi et al. (2024) | 2.216 | 1.846 |
| MOOSE-Chem | **3.686** | 2.443 |
| w/o multi-step | 3.588 | 2.529 |
| w/o multi-step & EU | 2.902 | 2.631 |

Table 14: Experiments and ablation study. The Matched Score is evaluated by `Gemini-1.5-Pro`.

| MS threshold | only non-EU branch | only EU branches | only EU-recombination branch |
|---|---|---|---|
| 5 | 16 | 46 | 20 |
| 4 | 19 | 54 | 24 |

Table 15: Number of hypotheses receiving high Matched Score (MS) from only non-EU branch, only EU branches, and only EU-recombination branch. Only the hypotheses with a MS that is higher than the MS threshold are counted.

|  | 5 | 4 | 3 | 2 | 1 | 0 |
|---|---|---|---|---|---|---|
|  | w/ significance feedback | | | | | |
| Average MS | 4 | 19 | 15 | 10 | 3 | 0 |
| Top MS | 33 | 7 | 10 | 1 | 0 | 0 |
|  | w/o significance feedback | | | | | |
| Average MS | 8 | 28 | 11 | 3 | 1 | 0 |
| Top MS | 34 | 13 | 4 | 0 | 0 | 0 |

Table 16: Effect of significance feedback (evaluated by `Claude-3.5-Sonnet`).

|  | Overall | Validness | Novelty | Significance | Potential |
|---|---|---|---|---|---|
| Average Rank Ratio | 0.65 | 0.75 | 0.76 | 0.73 | 0.70 |

Table 17: Average rank ratio ($\downarrow$) of the ground truth hypotheses (mixed with generated hypotheses)

## A.6   More Analysis on EU

Table 15 shows the number of hypotheses receiving high Matched Score from only non-EU branch, only EU branches, and only EU-recombination branch. Here, only non-EU branch can be seen as the hypotheses obtained directly without mutations. The hypotheses are from the same experiment in Table 10.

The result indicates that about one-third of high-quality hypotheses can be obtained directly without mutations. In addition, the recombination branch contains more high-quality hypotheses than the only non-EU branch.

## A.7   Effect of Significance Feedback

Table 16 presents an ablation study on the significance feedback. The results with significance feedback are from Table 12.

The results indicate that not using significance feedback can even lead to a better performance in terms of the Matched Score metric. We attribute this phenomenon to LLM's ability on creativity: when asked to generate significant hypotheses, LLMs tend to be more deviate from the existing information for more possible significance, resulting in a lower matched score. However, we should note that the matched score only measures the matching degree of one given ground truth hypothesis, and it is possible that the more deviated one is more significant.

## A.8   Ranking of Ground Truth Hypotheses

Intuitively if we rank the original hypothesis with the generated hypothesis, the original hypothesis may be ranked at the top for most of the time. But is it?

Table 17 shows the result, where we assign each ground truth hypothesis with a reward value $R(h)$ (in terms of validness, novelty, significance, and potential), and calculate its average rank ratio regarding the framework-generated hypotheses.

Surprisingly, the ground truth hypotheses are not ranked to the top. There are three possible reasons:

1. LLM does poorly on ranking hypotheses;

2. The generated hypotheses tend to describe their novelty and significance (although they are prompted to not to), which might influence the judgment;

3. The generated hypotheses may surpass the original in quality.

4. The generated hypotheses may sometimes have more details than the ground truth one (since the iterative usage of clarity feedback and refinement).

| Strict Background | Background Survey | Hit Ratio (top 20%) | Hit Ratio (top 4%) | Hit Ratio (top 0.8%) |
|---|---|---|---|---|
| ✓ | ✓ | 96.7% | 83.7% | 60.8% |
| ✓ | ✗ | 95.1% | 77.8% | 54.2% |
| ✗ | ✓ | 96.7% | 80.1% | 57.8% |

Table 18: Ablation table on background options for $Q1$. The corpus size is 300. For each screen window of 15 papers, 3 papers are selected.

### A.9 INFLUENCE OF RESEARCH BACKGROUND OPTIONS TO INSPIRATION RETRIEVAL

Table 18 shows the ablation study in terms of whether to use strict background (discussed in § 3) or survey or not. It indicates that a survey can largely help with the inspiration retrieval process. Surprisingly, without a strict background, the Hit Ratio goes down a bit. We attribute it to the reason that mentioning information related to the inspiration will discourage retrieving that inspiration, since in the prompt, we ask LLMs to search for inspirations, and the demonstration example indicates that inspirations should not be too similar to the background (to bring in additional information).

### A.10 DISCUSSION ON HALLUCINATION AND SCIENTIFIC DISCOVERY

In contrast to the traditional understanding that hallucination is purely a bad thing, LLM's scientific discovery ability in fact counts on its hallucination ability to find novel hypotheses: a novel hypothesis would not have been observed by itself, therefore all novel hypotheses come from the class of hallucination.

In essence, the research development of LLMs for automated scientific hypothesis discovery is to develop how to better leverage LLMs to hallucinate an unseen hypothesis that has more possibility to be valid.

### A.11 OTHER RELATED WORKS

#### A.11.1 REASONING

Scientific discovery is highly related to reasoning, since it requires a set of very complex reasoning processes to lead to new discovery.

Inductive reasoning (Yang et al., 2024a) is the most relevant reasoning type. It is about finding rules or hypotheses from observations. Scientific discovery is naturally an ultimate goal of inductive reasoning.

Inductive reasoning is a sub-reasoning type of logical reasoning. The other two sub-reasoning types are deductive reasoning (Clark et al., 2020) and abductive reasoning (Bhagavatula et al., 2020). Yang et al. (2023b) discuss their definitions and differences in detail.

Another relevant reasoning type is commonsense reasoning (Yang et al., 2020; 2023a). Scientific discovery can be seen as an opposite task, which is to reason far outside of commonsense, even to discover unknown knowledge.

#### A.11.2 RETRIEVAL

The retrieval of inspiration is a retrieval task, and RAG (Lewis et al., 2020) also works on retrieval. The main difference is that the current RAG method would most likely retrieve the information that is semantically the most similar to the input information (research background), while here our goal is to retrieve those information that was not known to be related to the input information before, but in fact is related. We assume that LLMs might have the ability to do it.

#### A.11.3 SELF CONSISTENCY

Self-consistency (Wang et al., 2023; Chen et al., 2023) might have a similar looking to the "evolutionary unit" (EU), as they all have expansion to several branches, and finally collect these branches into one.

A key difference is that EU is to explore more diverse options to choose the optimal one, while self-consistency is to find consistent voting between options.

## A.12 PROMPT TO GPT-4O FOR MATCHED SCORE

You are helping to evaluate the quality of a proposed research hypothesis in Chemistry by a phd student. The ground truth hypothesis will also be provided to compare. Here, we mainly focus on whether the proposed hypothesis has covered the key points in terms of the methodology in the ground truth hypothesis. You will also be given a summary of the key points in the methodology of the ground truth hypothesis for reference. Please note that for the proposed hypothesis to cover one key point, it is not necessary to explicitly mention the name of the key point, but might also can integrate the key point implicitly in the proposed method. The evaluation criteria is called 'Matched score', which is in a 6-point Likert scale (from 5 to 0). Particularly, 5 points mean that the proposed hypothesis (1) covers all the key points and leverage them similarly as in the methodology of the ground truth hypothesis, and (2) does not contain any extra key point that has apparent flaws; 4 points mean that the proposed hypothesis (1) covers all the key points (or at least three key points) and leverage them similarly as in the methodology of the ground truth hypothesis, (2) but also with extra key points that have apparent flaws; 3 points mean that the proposed hypothesis (1) covers at least two key points and leverage them similarly as in the methodology of the ground truth hypothesis, (2) but does not cover all key points in the ground truth hypothesis, (3) might or might not contain extra key points; 2 points mean that the proposed hypothesis (1) covers at least one key point in the methodology of the ground truth hypothesis, and leverage it similarly as in the methodology of ground truth hypothesis, (2) but does not cover all key points in the ground truth hypothesis, and (3) might or might not contain extra key points; 1 point means that the proposed hypothesis (1) covers at least one key point in the methodology of the ground truth hypothesis, (2) but is used differently as in the methodology of ground truth hypothesis, and (3) might or might not contain extra key points; 0 point means that the proposed hypothesis does not cover any key point in the methodology of the ground truth hypothesis at all. Please note that the total number of key points in the ground truth hypothesis might be less than three, so that multiple points can be given. E.g., there's only one key point in the ground truth hypothesis, and the proposed hypothesis covers the one key point, it's possible to give 2 points, 4 points, and 5 points. In this case, we should choose score from 4 points and 5 points, depending on the existence and quality of extra key points. 'Leveraging a key point similarly as in the methodology of the ground truth hypothesis' means that in the proposed hypothesis, the same (or very related) concept (key point) is used in a similar way with a similar goal compared to the ground truth hypothesis (not necessarily for the proposed hypothesis to be exactly the same with the groudtruth hypothesis to be classified as 'similar'). When judging whether an extra key point has apparent flaws, you should use your own knowledge to judge, but rather than to rely on the count number of pieces of extra key point to judge.

Please evaluate the proposed hypothesis based on the ground truth hypothesis.
The proposed hypothesis is:
The ground truth hypothesis is:
The key points in the ground truth hypothesis are:
Please evaluate the proposed hypothesis based on the ground truth hypothesis, and give a score.

## A.13 GENERATED HYPOTHESES WITH LOW MATCHED SCORE ARE NOT NECESSARILY BAD

MS only measures the similarity between the generated $h$ and the ground truth $h$. Receiving an MS as 0 or 1 does not mean the generated $h$ is bad. Only real lab experiments can check each $h$.

## A.14 EVALUATION AGREEMENT BETWEEN EXPERT EVALUATION AND GPT-4O EVALUATION

Table 19 shows the agreement between expert evaluation and automatic evaluation (by GPT-4o) on MS. Hard consistency is assigned to 1 only if the two scores are exactly the same, else is assigned to 0. Soft consistency is assigned to 1 only if the absolute difference between the two scores is less than 2, else is assigned to 0.

| #Comparison Pairs | Hard Consistency Score | Soft Consistency Score |
|:---:|:---:|:---:|
| 392 | 0.345 | 0.542 |

Table 19: Consistency score between expert evaluation and `GPT-4o` evaluation.

| #Comparison Pairs | Hard Consistency Score | Soft Consistency Score |
|:---:|:---:|:---:|
| 48 | 0.438 | 0.854 |

Table 20: Consistency score between experts in expert evaluation.

The results show a medium to high consistency between expert evaluation and automatic evaluation. The main reason is that, in practice, the automatic evaluation is usually 1 to 2 points higher than expert evaluation, since GPT-4o can usually find a way to explain how the generated hypothesis is related to the ground truth hypothesis in terms of the main innovations. While this explanation usually is not wrong, the experts might find that compared to the MS given by GPT-4o, the generated hypotheses might not be clear enough to deliver the new innovation information to scientists when using it as a copilot advice.

### A.15 AGREEMENT BETWEEN EXPERT EVALUATION

Table 20 shows the consistency score between experts. It is conducted by having the third chemistry expert check the randomly sampled expert evaluation from the previous two chemistry experts. Hard consistency is assigned to 1 only if the two scores are exactly the same, else is assigned to 0. Soft consistency is assigned to 1 only if the absolute difference between the two scores is less than 2, else is assigned to 0.

The results show a high consistency between experts, indicating a high reliability of reference-based expert evaluation.

### A.16 CASE STUDIES

#### A.16.1 CASE 1 (MS BY EXPERT: 3)

Ground truth $h$ (Bu et al., 2024): *The main hypothesis is that a **nitrogen-doped ruthenium (Ru) electrode** can effectively catalyze the reductive deuteration of (hetero)arenes in the presence of $\mathbf{D_2O}$, leading to high deuterium incorporation into the resulting saturated cyclic compounds. The findings validate this hypothesis by demonstrating that this electrocatalytic method is highly efficient, scalable, and versatile, suitable for a wide range of substrates.*

Generated $h$: *A pioneering integrated electrocatalytic system leveraging **ruthenium** nanoparticles embedded in **nitrogen-doped** graphene, combined with a dual palladium-coated ion-exchange membrane reactor, will catalyze efficient, scalable, and site-selective reductive deuteration of aromatic hydrocarbons and heteroarenes. Utilizing deuterium sources from both $D_2$ gas and $\mathbf{D_2O}$, this system will optimize parameters through real-time machine learning-driven dynamic adjustments. Specific configurations include ruthenium nanoparticle sizes (2-4 nm), nitrogen doping levels (12-14%), precisely engineered palladium membranes (5 micrometers, ensuring 98% deuterium-selective permeability), and advanced cyclic voltammetry protocols (1-5 Hz, -0.5V to -1.5V).*

Expert's analysis: *The proposed hypothesis effectively covers two key points from the ground truth hypothesis: **the incorporation of ruthenium (Ru) and the use of $\mathbf{D_2O}$ as a deuterium source** within the electrocatalytic system. However, the current content does not detail the mechanism by which Ru-D is produced, which is essential for explaining the process of reductive deuteration. Nevertheless, the results are still insightful. The specific level of nitrogen doping, for example, is highly suggestive and warrants further investigation. Overall, the match remains strong in its alignment with the original hypothesis while also presenting opportunities for deeper exploration.*

### A.16.2 CASE 2 (MS BY EXPERT: 4)

Ground truth $h$ (Wang et al., 2024a): *By integrating guanidine sulfate $(Gdm)_2SO_4$ into a poly vinyl alcohol (PVA) hydrogel and employing directional freezing to create aligned channels, it is possible to achieve a flexible thermogalvanic armor (FTGA) with a Carnot-relative efficiency exceeding 8% while maintaining high mechanical strength. This integration allows for enhanced thermopower and mechanical robustness, exceeding the performance of traditional quasi-solid thermocells.*

Generated $h$: *Leveraging an integrated multi-layer hydrogel-ion assembly, constructed through advanced freeze-casting, salting-out processes, and bioinformatics-driven macromolecular design, will enhance the Carnot-relative efficiency and mechanical robustness of flexible thermogalvanic devices. This approach reimagines the use of poly(vinyl alcohol) hydrogels and incorporates novel ion-specific interactions informed by the Hofmeister series with uniquely potent ions like magnesium and lithium. Precision ion concentrations (0.15 to 0.45 M) are optimized to amplify thermoelectric responses and ionic conductance. Molecular dynamics simulations, employing detailed quantum chemistry models, will validate the enhanced ionic interactions and entropy modulation. Comprehensive characterization through nanoindentation, electron microscopy, and advanced computational platforms will verify crystalline polymer structures and stratified ionic gel configurations. By setting new benchmarks in efficiency and flexibility, this innovation is poised to revolutionize energy solutions in wearable technologies, outperforming current models in both scalability and application feasibility.*

Expert's Analysis:

*The proposed hypothesis effectively covers key points, mirroring the ground truth hypothesis. It incorporates the Hofmeister series by mentioning "ion-specific interactions informed by the Hofmeister series," which aligns with the ground truth's reference to "guanidine sulfate $(Gdm)_2SO_4$" in the hydrogel integration, acknowledging that sulfate ions are a significant component of the Hofmeister series. Additionally, the proposed hypothesis employs "freeze-casting," which reflects the ground truth's "directional freezing" to create structured channels within the hydrogel. This approach enhances both mechanical strength and thermopower—objectives that are consistent with those of the ground truth hypothesis.*

*Moreover, the inclusion of salting-out processes in the proposed hypothesis does indeed contribute to improving the mechanical properties of thermoelectric materials. However, the selection of magnesium and lithium as specific examples is problematic. Overall, the proposed hypothesis provides valuable insights and serves as a source of inspiration for further exploration within this domain.*

### A.16.3 CASE 3 (MS BY EXPERT: 3)

Ground truth $h$ (Shibahara et al., 2024): *Rh(I)-catalyzed vinylidene formation and enantioselective control via chiral diphosphine ligands can work together to enable a selective [2+2+1] cycloaddition reaction using three different $2\pi$-components. The formation of the vinylidene intermediate ensures reaction selectivity, while the chiral ligands, through their dihedral angles, control the enantiomeric outcome. Together, these factors will result in the highly enantioselective synthesis of chiral 3-methylenecyclopent-1-ene derivatives.*

Generated $h$: *An enantioselective [2+2+1] cycloaddition involving cycloalkenes, acetylenecarboxylates, and terminal alkynes can be achieved using a dynamically tunable Rh(I) pincer complex system, specifically incorporating PNP and PONOP ligands, to control the alkyne-to-vinylidene transformation and enhance selectivity towards chiral 3-methylenecyclopent-1-ene derivatives. By adjusting the P–Rh–P bite angle with novel ligand modifications and utilizing a blended solvent system containing toluene and dichloromethane at controlled temperatures ($25°C$ to $80°C$), reaction conditions can be fine-tuned to suppress side reactions. Computational methods (DFT, QTAIM, and NBO) will guide the optimization process, supported by real-time monitoring using FTIR and NMR. Exploring dual-ligand strategies, including phosphine/phosphinite combinations, will offer insights into previously unreported mechanistic pathways, ensuring a reproducible and scalable synthesis method applicable to pharmaceuticals and advanced material synthesis.*

Expert's Analysis: *The hypothesis generated in Case 2 receives a score of 3, reflecting its accurate identification of the metal and ligand necessary for the catalytic process, as well as its articulation of the mechanism underlying chiral control via ligand geometry. However, the inclusion of mixed*

*solvents, which were not utilized in the actual study, prevents a higher rating. This prediction successfully asserts the necessity of rhodium metal and a phosphine ligand, while also proposing the significance of the chelating angle between the phosphine ligand and rhodium, in alignment with the hypothesis presented in the referenced article. Furthermore, although the DFT calculations mentioned in the hypothesis are not addressed in the article, they represent a promising avenue for future research in organic chemistry.*

### A.16.4 CASE 4 (MS BY EXPERT: 3)

Ground truth $h$ (Suzuki et al., 2024): *The hypothesis is that phosphorus ylides, when exposed to photoredox catalysis, can undergo single-electron oxidation to generate radical cations, which engage in radical-polar crossover reactions. These intermediates can then sequentially form C–C and C=C bonds through a formal cycloaddition process, offering a powerful method for constructing six-membered carbocycles from simple substrates.*

Generated $h$: *A sophisticated dual-modal photoredox strategy synergistically integrates phosphorus ylides with a synchronized dual-wavelength excitation (450 nm and 520 nm) system to incite a regioselective three-component cycloaddition. This method exploits both net-neutral radical/polar crossover pathways and metal-free carbyne generation to convert inert C–H and C=P bonds into C–C and C=C bonds efficiently. Implemented within an optimized mixed solvent system (acetone/acetonitrile) under an ambient nitrogen atmosphere, the approach utilizes precise kinetic modulation and catalytic control, with computational validation (employing DFT models) underpinning mechanistic insights and regioselectivity. Experimental verification through in situ spectroscopic techniques (e.g., UV-Vis and NMR) enhances synchronization precision, curtailing side reactions. This methodology pledges substantial advancements in sustainable pharmaceutical synthesis and innovative polymer architecture formation, specifically by improving production efficiency for complex molecules and novel materials.*

Expert's Analysis: *The generated hypothesis also merits a score of 3, as it correctly anticipates the use of photocatalysis and highlights the significant influence of solvent on the reaction. However, since dual wavelength catalysis and solvent mixing were not employed in the actual experiment, a higher score is not warranted. Notably, despite the proposed mixed solvents not being used in the study, their composition comprises the two best-performing single solvents from the actual research, thus providing valuable insights that remain relevant to the ongoing investigation.*

