# OpenReview forum: "MOOSE-Chem: Large Language Models for Rediscovering Unseen Chemistry Scientific Hypotheses"
_ICLR.cc/2025/Conference — ICLR 2025 Poster_

### Official Review · Reviewer_km2P · 2024-10-31

**Soundness:** 3
**Presentation:** 2
**Contribution:** 2
**Rating:** 6
**Confidence:** 4

**Summary:**

This paper explores the potential of Large Language Models (LLMs) to generate novel, valid hypotheses in the chemistry domain by employing a multi-agent framework. It targets the rediscovery of scientific hypotheses using LLMs based on recent high-impact chemistry publications (since 2024), using a framework that separates tasks into three stages: (1) finding "inspiration" papers from a predefined literature corpus, (2) generating hypotheses by associating these inspirations with a background question, and (3) ranking hypotheses based on quality. To solve this, the authors proposed a multi-agent framework existing framework with an evolutionary algorithm to iteratively mutate hypotheses based on reward feedback (validness, novelty, clarity, and significance).

**Strengths:**

1. The use of LLMs for scientific hypothesis generation is a growing area, and extending this work to complex fields like chemistry holds practical importance if achieved meaningfully.
2. The use of beam search and evolutionary algorithms for hypothesis refinement seems simple and powerful, as it systematically explores multiple possibilities.
3. The framework does not require expensive training or fine-tuning.

**Weaknesses:**

1. There is a key fundamental assumption that the authors did not discuss here: reliance on general-purpose LLMs containing implicit domain knowledge. While this may be reasonable for powerful, large-scale models like GPT-4 or domains like social sciences, it is overly strong and may not apply to smaller or open-source models in this case. A deeper analysis (maybe more experiments) of how this assumption impacts performance across other LLMs (apart from GPT-4 or closed-source ones) is needed.
2. The assumption that LLMs have sufficient knowledge to connect novel and unrecognized scientific relations without any domain-specific tuning seems too bold. The model's effectiveness may degrade in more niche areas or specific research questions.
3. Much of this framework — multi-agent, hypothesis scoring, and the background-inspiration format closely resembles existing work in this area. Given the limited scale of benchmarking and experiments in this paper, the technical contribution seems limited for this venue.
3. The selection of inspiration papers seems unclear and might not align with typical challenges or open research questions in chemistry. A more detailed explanation of the selection process of those 51 papers and corresponding inspiration papers, including how relevance, significance, and novelty are ensured, is needed.
3. The paper presentation could be improved, particularly in the introduction and methodology. The technical details regarding probabilistic inference seem unnecessary.

**Questions:**

1. Can the authors clarify how/whether MOOSE-Chem substantially differs from previous work in multi-agent scientific discovery systems?
2. How would the authors envision MOOSE-Chem supporting a chemist in hypothesis generation beyond rediscovering existing knowledge? Have any of the generated hypotheses been practically tested, or do they align with ongoing chemistry research needs?

---

> ### Author Response · Authors · 2024-11-20
> **Reply to Reviewer km2P - Part 1/2**
>
> Thank you for your insightful feedback. Before addressing your questions directly, allow us to briefly summarize the key contributions of our paper.
>
>
> Overall, our contributions are:
> 1. We provide the first mathematical derivation on how to decompose the seemingly-impossible-to-solve question P(hypothesis | research background) into many executable and practical smaller steps. This decomposition make P(hypothesis | research background) possible to be practical;
> 2. We develop a scientific discovery framework directly based on the mathematical derivation. Different from previous work, we propose an evolutionary algorithm based method to better associate background and inspiration, multi-step inspiration retrieval and composition, and an efficient ranking method. In addition, the framework can be applied to chemistry and material science, where previous methods only covered social science, NLP, and biomedical domains;
> 3. We construct a benchmark by three chemistry PhD students, consisting of 51 chemistry papers published on Nature, Science, or a similar level, decomposing each paper into the research background, inspirations, and hypothesis;
> 4. For the first time, we show that an LLM-based framework can largely rediscover chemistry hypotheses that have published in Nature and Science. It is guaranteed that the rediscovery is not because of data contamination, because we have controlled the date of the training corpus of the LLM and the date of the chemistry papers that are available online.
>
>
> **Q1**: This paper relies on LLMs for the framework. However, it is unclear on how smaller language models perform on this task.
>
> **A1**: Indeed, additional analysis on the scale of LLMs would make this paper more informative.
>
> Here we have tested Llama-3.1-8b,  Llama-3.1-70b, and Llama-3.1-405b on their performance on inspiration retrieval. The results are averaged across the 51 benchmark papers:
>
> | Model          | Hit Ratio (top 0.8%) | Hit Ratio (top 4%) | Hit Ratio (top 20%) |
> |----------------|----------------------|--------------------|---------------------|
> | llama-3.1-8B   | 0.268                | 0.435              | 0.716               |
> | llama-3.1-70B  | 0.500                | 0.778              | 0.880               |
> | llama-3.1-405B | 0.528                | 0.741              | 0.926               |
> | GPT-4o         | 0.608                | 0.837              | 0.967               |
>
> The results indicate a trend, where in general larger LLMs have a better heuristics on inspiration retrieval.
>
>
>
>
> **Q2**: The assumption that LLMs have sufficient knowledge to connect novel and unrecognized scientific relations without any domain-specific tuning seems too bold.
>
> **A2**:
> (1) We claim that it is an assumption, but not an argument (line 257);
> (2) Based on the experimental results, our conclusion regarding the assumption is that “it is possible to be true” (line 349~351).
>
> We think proposing a counter-intuitive assumption that is consistent with all of the experimental observations should be a good contribution to the field.
>
> We will make our conclusion to the assumption directly next to the proposal of the assumption, which should avoid the misunderstanding.
>
>
> **Q3**: Much of this framework — multi-agent, hypothesis scoring, and the background-inspiration format closely resembles existing work in this area.  The technical contribution seems limited for this venue.
>
> **A3**: Sorry for the misunderstanding. We have summarized the four contributions in this paper (as shown above). We will add them to the end of the introduction section to avoid misunderstanding.
>
>
> **Q4**: A more detailed explanation of the inspiration selection process for the benchmark papers is needed.
>
> **A4**: Sure. For each of the 51 papers in the benchmark, its inspiration is usually stated in the introduction section. The chemistry PhD students find (or select) the inspiration paper based on the reference in the introduction section.
>
> Specifically, an inspiration paper is usually followed with “Motivated by xxx [1]”. Here [1] would be one of the inspiration papers.

---

> ### Author Response · Authors · 2024-11-20
> **Reply to Reviewer km2P - Part 2/2**
>
> **Q5**: The technical details regarding probabilistic inference seem unnecessary.
>
> **A5**: Sorry for the misunderstanding.  They are not probabilistic inference, but the decomposition of the ultimate function of scientific discovery: P(hypothesis | research background). The derivation/proof directly supports the design of the framework. It corresponds to the first contribution in the summarized four contributions. We will make it more clear in the paper.
>
>
> **Q6**: Can the authors clarify how/whether MOOSE-Chem substantially differs from previous work in multi-agent scientific discovery systems?
>
> **A6**: Yes, please refer to A3.
>
>
> **Q7**: How would the authors envision MOOSE-Chem supporting a chemist in hypothesis generation beyond rediscovering existing knowledge? Have any of the generated hypotheses been practically tested, or do they align with ongoing chemistry research needs?
>
> **A7**: Yes, and we have simulated a real situation to support a chemist. It can be found in section 6, line 460~461, where we quote it here: “Here we perform experiments in a setting similar to the copilot in-the-wild setting.”. The “copilot in-the-wild setting” represents the real situations to support chemists.
>
> Specifically, the MOOSE-Chem framework only needs to take three inputs: research question, background survey, and lots of high quality random literature. A chemist only needs to prepare their research question and background survey. The literature can be randomly papers collected in advance. They MOOSE-Chem will start to inference to a ranking of hypotheses.
>
> We show that MOOSE-Chem have rediscovered several chemistry hypotheses that were published in Nature and Science in 2024 (and we have made sure that it is not because of data contamination by splitting the date of the published papers and the date of the pretraining corpus of the LLM we use). These hypotheses surely have been tested.
>
>
>
> **We greatly appreciate your comprehensive feedback. We hope that our responses have satisfactorily addressed all your queries. Should you have further questions or suggestions for enhancing our manuscript, we warmly welcome your input.**

---

> > ### Comment · Reviewer_km2P · 2024-11-21
> > **Response to rebuttal**
> >
> > Thanks for the rebuttal and for clarifying the contributions. The additional experiments are very helpful to put this work in context. I have raised the score to 6. Good luck!

---

> ### Author Response · Authors · 2024-11-22
> **Response to reviewer**
>
> Thank you for your thoughtful review, reconsideration, and the kind wishes! Glad to know that your concerns have been addressed!

---

### Official Review · Reviewer_TMW5 · 2024-11-03

**Soundness:** 4
**Presentation:** 4
**Contribution:** 3
**Rating:** 8
**Confidence:** 4

**Summary:**

This paper introduces a framework for scientific hypothesis discovery
and generation using large language models (LLMs). The authors gathered
a carefully curated dataset of scientific papers where questions and hypotheses were extracted for evaluation. The authors propose "MOOSE-CHEM", a
method based on previous work "MOOSE" but with modifications
to adapt it for chemistry literature and that improve Overall
performance (refinement, mutation). The framework is also formally motivated.
The proposed approach is evaluated in each of the main three tasks
that compose it: P (i|b), P (h|b, i), and R(h). Overall the results show great
performance in the tasks with GPT-4o, with evaluations that include both
experts and GPT-4o, and where various hyperparameters are tested (e.g. corpus size).

**Strengths:**

The paper is generally well-written, and in good English. The text is clear and
the authors did a good job guiding the reader through the motivation,
the derivation of the method and motivating each of the proposed steps
and experiments. The topic of the paper is very relevant and the results
are positive. Related work is well covered, and experiments are included
that compare the proposed method with previous work. Every claim made
on the performance of the method is generally backed up with experiments.
I think the paper is generally a great and novel contribution to the field.

**Weaknesses:**

My main concerns with the paper are regarding the reproducibility,
clarity and discussion of the approach:

- Reproducibility: we note that the authors introduce a scientific benchmark,
along with a novel framework for hypothesis generation. However,
the authors do not provide access to the novel-introduced benchmark,
which hampers the ability to really discriminate the difficulty of the tasks
at hand. Additionally, this impedes the ability to reproduce the results or
for future work to compare the performance with the proposed method.
I would suggest the authors to make the benchmark data available if possible.
Similarly, no access to the source code is provided. While available source code is not
a requirement, for a method that relies heavily on LLMs prompting, it would
be very beneficial to at least have access to the prompts used in the
work, just for the sake of reproducibility.

- (Related to the previous issue): There are parts of the paper that lack
a formal definition, which again, hampers the
ability to reproduce the work. For instance, the authors mention that they
introduce a novel "Evolutionary Algorithm (EA)" that mutates, refines and
recombines hypotheses, however, it is unclear how the mechanisms of
mutation, refinement and recombination are implemented. I would suggest
the authors to provide, when possible more technical details on the
proposed method.

- While the work includes a fair comparison with previous work, the work
doesn't include any discussion on the limitations of the proposed method,
nor any future work. It would be very much appreciated if the authors
could provide a discussion on the limitations of the proposed method
to guide potential future work in the field.

**Questions:**

This includes a list of minor issues, questions and suggestions:

- Through the paper, it is mentioned that the inspiration shouldn't be related
to the given background. Is this enforced in the method, or is this verified
somehow?

- The proposed method of Evolutionary Algorithm (EA) sounds to be related
to other methods in LLMs that try to improve the consistency/quality of
the answers, e.g.
  - Self-Consistency (SC) (Wang et al., "Self-Consistency Improves Chain of Thought Reasoning in Language Models")
  - Universal SC (Chen et al. "Universal Self-Consistency for Large Language Model Generation")
  - Multiple Chain of Thought (MCT), Self reflection and/or self-correction approaches, etc. I thus believe that some related work could be included regarding this algorithm in the paper.


- On line 351, claim that "is possible to be true" that LLMs are able to create
novel knowledge. While the authors are correct here since it is not clear
whether this is true or not, could the authors provide other explanations
on why the LLMs are able to correctly find inspiration papers?


- Please if possible provide some description of how P (i|b), P (h|b, i), and R(h)
inputs and outputs look like. From the example provided, I can guess that
hypothesis is one paragraph long, but it would be great for the reader
to have this stated clearly.

- I honestly don't fully understand the experimental setting reported in section 5.1.
What are the "ground truth inspiration papers" and how are these related to
the 51 papers from the benchmark? In Table 3, does this mean that when a corpus size
of 150 is used, 120 papers are considered ground truth?

- From Table 3, it almost seems like corpus size doesn't have much effect on the Hit ratio,
is this correct? If so, could you elaborate more in this regard?

- If I got the idea right, in the results of Table 1, if a screen window size is used, the corpus is 300 and 3 papers are selected for each window, then round 1 has (300/10)*3 = 90, round 2 has
(90/10)*3 = 27, and so on, is this correct? It would be more clarifying to include the amount of papers in each round in the table or in a separate table. It would be also clarifying
to include a baseline comparison of the performance based on pure random screening (this is, select papers randomly).

- From the results reported in Table 7, it seems that the background survey (almost) doesn't have any
effect on MS, however, this is somehow counterintuitive. Shouldn't the background survey improve performance as it happens in Q1?

- I would suggest the authors format the rounds in Table 4 to go along
with the direction of the text, this is, from left to right.

- Could you please clarify why there are NAs in Tables 3 and 4?

- On line 134: "contrusted by multiple chemistry PhD students".
For clarity, it would be great to know how many students.

- While it is not mandatory, given the stochasticity and prone-to-hallucinations nature of LLMs,
it is a good practice to use countermeasures to reduce hallucinations, e.g. sampling multiple times with majority voting,
or using different models for comparison. This is not an issue, but it would be
interesting to see how much variability the results reported in the experiments
are or if the method is robust to hallucinations.

---

> ### Author Response · Authors · 2024-11-20
> **Reply to Reviewer TMW5 - Part 1/3**
>
> We are grateful for the detailed comments. Here, we present our responses corresponding to the questions you've asked.
>
>
> **Q1**: Reproducibility: I would suggest the authors to make the benchmark data available if possible. Similarly, no access to the source code is provided.
>
> **A1**: Yes, we agree. All the benchmark data and code can be found in https://anonymous.4open.science/r/MOOSE-Chem. The data and code are also available on github since early October.
>
> You are right, we should have added an anonymous link into the submission. We will add the github link to the paper.
>
>
> **Q2**: It is unclear how the mechanisms of mutation, refinement and recombination are implemented.
>
> **A2**: Thank you for the feedback. Originally we describe it in line 279~287. We add more details to it and modify it as below:
>
> Specifically, given b and i, the framework will try multiple hypothesis “mutations” m, where each m is a unique way to associate b and i together. Then we further develop each m independently by providing feedback to each m in terms of validity, novelty, clarity, and significance, and then refine them based on the feedback. We assume the refined hypothesis should be in better quality, so that the refined hypothesis is “selected”, while the previous hypothesis is “eliminated” by the “environment”. Finally the framework “recombine” the remaining selected m, leveraging the advantages from every m to propose h to better associate b and i. The recombine step is implemented as given b, i, and several remaining selected hypotheses m as the input, and instruct LLMs to compose a final hypothesis that keeps the good parts from each m while discarding the low-quality parts.
>
> Please let us know if there’s any place you would like us to make more justifications. We would add more justifications at once.
>
>
> **Q3**: The work doesn't include any discussion on the limitations of the proposed method, nor any future work.
>
> **A3**: ok, sure. One important future work which could largely improve the validity of the generated hypotheses from the proposed method is automation on experiment implementation. With it, the validity of the generated hypothesis can be guaranteed and can result in a smaller hypothesis recommendation range to researchers (where many noisy hypotheses are filtered). The automation on experiment implementation includes robotics (for chemistry experiments) and code implementation (for CS experiments)
>
> Another future research direction is how to improve language model’s ability on scientific discovery. It is a largely unexplored research field.
>
>
> **Q4**: It is mentioned that the inspiration shouldn't be related to the given background. Is this enforced in the method, or is this verified somehow?
>
>
> **A4**: It is a theoretical guideline from the philosophy of science community. In practice, we incorporate it in the prompt of the inspiration retrieval module to encourage it.
>
>
> **Q5**: The proposed method of Evolutionary Algorithm (EA) sounds to be related to other methods in LLMs that try to improve the consistency/quality of the answers. I thus believe that some related work could be included regarding this algorithm in the paper.
>
> **A5**: We will discuss the related work you mentioned. A key difference is that EA is to explore more diverse options to choose the optimal one, but not to find consistency between options.
>
> **Q6**: On line 351, claim that "is possible to be true" that LLMs are able to create novel knowledge. While the authors are correct here since it is not clear whether this is true or not, could the authors provide other explanations on why the LLMs are able to correctly find inspiration papers?
>
> **A6**: A short answer is, after lots of pretraining seeing lots of description of knowledge, LLMs have the heuristics (just like human researchers) on finding inspiration papers.
>
> Line 260~266 also discusses this question, we quote as below:
>
> “It might not be too bold, since Tshitoyan et al. (2019) have shown that word embedding obtained from unsupervised learning on 3.3 million scientific abstracts (that are related to material science) can recommend materials for functional applications several years before their discovery. Here the functional applications can be seen as b, and the recommended materials can be seen as i, or even be directly seen as h, if the recommended material is exactly the one to be discovered later. It probably indicates that LLMs trained with significantly more literature tokens and significantly more parameters, might already be able to identify the relation between lots of knowledge pairs that is unknown to be related by any scientist.”

---

> ### Author Response · Authors · 2024-11-20
> **Reply to Reviewer TMW5 - Part 2/3**
>
> **Q7**: Please if possible provide some description of how P (i|b), P (h|b, i), and R(h) inputs and outputs look like. From the example provided, I can guess that hypothesis is one paragraph long, but it would be great for the reader to have this stated clearly.
>
> **A7**: Sure.
>
> **P(i|b)’s input prompt**:
>
> ["You are helping with the scientific hypotheses generation process. Given a research question, the background and some of the existing methods for this research question, and several top-tier publications (including their title and abstract), try to identify which publication can potentially serve as an inspiration for the background research question so that combining the research question and the inspiration in some way, a novel, valid, and significant research hypothesis can be formed. The inspiration does not need to be similar to the research question. In fact, probably only those inspirations that are distinct with the background research question, combined with the background research question, can lead to a impactful research hypothesis. The reason is that if the inspiration and the background research question are semantically similar enough, they are probably the same, and the inspiration might not provide any additional information to the system, which might lead to a result very similar to a situation that no inspiratrions are found. An example is the backpropagation of neural networks. In backpropagation, the research question is how to use data to automatically improve the parameters of a multi-layer logistic regression, the inspiration is the chain rule in mathematics, and the research hypothesis is the backpropagation itself. In their paper, the authors have conducted experiments to verify their hypothesis. Now try to select inspirations based on background research question. \nThe background research question is: ", "\n\nThe introduction of the previous methods is:", "\n\nThe potential inspiration candidates are: ", "\n\nNow you have seen the background research question, and many potential inspiration candidates. Please try to identify which three literature candidates are the most possible to serve as the inspiration to the background research question? Please name the title of the literature candidate, and also try to give your reasons. (response format: 'Title: \nReason: \nTitle: \nReason: \nTitle: \nReason: \n')"]
>
> **P(i|b)’s output**:
>
> Title: xxx\nReason: yyy\nTitle: xxx\nReason: yyy\nTitle: xxx\nReason: yyy\n
>
>
> **P (h|b, i)’s input prompt**:
>
> ["You are helping with the scientific hypotheses generation process. We in general split the period of conducting research into four steps. Firstly it's about finding a good and specific background research question, and an introduction of the previous methods under the same topic; Secondly its about finding inspiration (mostly from literatures), which combined with the background research question, can lead to a impactful research hypothesis; Thirdly it's hypothesis generation based on the background research question and found inspiration; Finally it's about designing and conducting experiments to verify hypothesis. An example is the backpropagation of neural networks. In backpropagation, the research question is how to use data to automatically improve the parameters of a multi-layer logistic regression, the inspiration is the chain rule in mathematics, and the research hypothesis is the backpropagation itself. In their paper, the authors have conducted experiments to verify their hypothesis. Now we have identified a good research question, and we have found a core inspiration in a literature for this research question. Please help us generate a novel, valid, and significant research hypothesis based on the background research question and the inspiration. \nThe background research question is: ", "\n\nThe introduction of the previous methods is:", "\n\nThe core inspiration is: ", "\n\nNow you have seen the background research question and the core inspiration. Please try to generate a novel, valid, and significant research hypothesis based on the background research question and the inspiration. (response format: 'Hypothesis: \nReasoning Process:\n')"]
>
> **P (h|b, i)’s output**:
>
> Hypothesis: xxx\nReasoning Process: yyy\n
>
>
> **R(h)**'s instruction can be found at Appendix A.3. It's a bit too long to copy them here. Please check Appendix A.3 if you are interested.

---

> ### Author Response · Authors · 2024-11-20
> **Reply to Reviewer TMW5 - Part 3/3**
>
> **Q8**: I honestly don't fully understand the experimental setting reported in section 5.1. What are the "ground truth inspiration papers" and how are these related to the 51 papers from the benchmark? In Table 3, does this mean that when a corpus size of 150 is used, 120 papers are considered ground truth?
>
> **A8**:For each paper of the 51 papers from the benchmark, it has two to three ground truth inspirations, where each inspiration corresponds to one inspiration paper. The total number of the ground truth inspiration papers for the full 51 papers in the benchmark is 120. As a result, for each paper out of 51 in the benchmark, only two to three papers are considered ground truth in a corpus size of 150.
>
>
> **Q9**: From Table 3, it almost seems like corpus size doesn't have much effect on the Hit ratio, is this correct? If so, could you elaborate more in this regard?
>
>
> **A9**: Yes it is correct. Because the Hit ratio is evaluated with a fixed selected ratio, e.g., if we only keep 20% of the corpus size, what is the hit ratio of the 20% remaining corpus. While larger corpus will keep a similar hit ratio, it also has more “noise” papers. The number of noise papers can be counted as 20% * |corpus| - |ground truth papers|.
>
>
> **Q10**: If I got the idea right, in the results of Table 1, if a screen window size is used, the corpus is 300 and 3 papers are selected for each window, then round 1 has (300/10)*3 = 90, round 2 has (90/10)*3 = 27, and so on, is this correct? It would be more clarifying to include the amount of papers in each round in the table or in a separate table. It would be also clarifying to include a baseline comparison of the performance based on pure random screening (this is, select papers randomly).
>
>
> **A10**: Yes, it is correct. It would be more clarifying the amount of papers. We will illustrate it in the paper, and add the pure random screening baseline.
>
> Specifically, the expectation of the random screening baseline is:
>
> | Hit Ratio (top 0.016%) | Hit Ratio (top 0.8%) | Hit Ratio (top 4%) | Hit Ratio (top 20%) |
> |--|---|-----|----|
> |                  0.016 |                 0.08 |               0.04 |                 0.2 |
>
> **Q11**: From the results reported in Table 7, it seems that the background survey (almost) doesn't have any effect on MS, however, this is somehow counterintuitive. Shouldn't the background survey improve performance as it happens in Q1?
>
> **A11**: It is a bit counterintuitive. The main reason is that we use a strict background survey for the experiments in Table 7.
>
> The strict version of background survey means the survey shouldn’t contain any information that is directly related to the hypothesis in any sense.
>
> The reason we add a strict version for background question and background survey in the benchmark is that many hypotheses are making relatively minor modifications based on existing methods covered by the survey, and the question can be very insightful to provide a hint on the general direction of the hypothesis. In practice, these situations are entirely possible, especially when the scientist users can provide a more comprehensive survey on existing methods. Here we also keep the strict version to make the task more challenging, and encourage developing methods to better assist scientists even when they are also new to their research topic.
>
>
> **Q12**: I would suggest the authors format the rounds in Table 4 to go along with the direction of the text, this is, from left to right.
>
> **A12**: Thank you for the suggestion. We will reverse the order. It is hard to understand why I have designed the table in this way after seeing your suggestion.
>
> **Q13**: Could you please clarify why there are NAs in Tables 3 and 4?
>
> **A13**: Yes, because when the corpus size is not too big and the remaining ratio is too low (e.g., 0.016%), the remaining inspiration will be less than 1. The result is not meaningful, therefore we use NA at the place.
>
> **Q14**: On line 134: "constructed by multiple chemistry PhD students". For clarity, it would be great to know how many students.
>
> **A14**: Three PhD students. We will make it clear in the paper.
>
> **Q15**: While it is not mandatory, given the stochasticity and prone-to-hallucinations nature of LLMs, it is a good practice to use countermeasures to reduce hallucinations, e.g. sampling multiple times with majority voting, or using different models for comparison. This is not an issue, but it would be interesting to see if the method is robust to hallucinations.
>
> **A15**: In fact, we count on LLM’s hallucination ability to find novel hypotheses: a novel hypothesis would not have been observed by the LLMs, therefore all novel hypotheses come from the class of hallucination.
>
>
> **We greatly appreciate your comprehensive feedback. We hope that our responses have satisfactorily addressed all your queries. Should you have further questions or suggestions for enhancing our manuscript, we warmly welcome your input.**

---

> ### Author Response · Authors · 2024-11-22
> **Looking forward to your reply**
>
> Dear Reviewer,
>
> Thank you for your insightful comments and suggestions. We have submitted responses to address the points you've raised, including clarifications, additional results, and anonymous github link to the full code and dataset. We hope that these will sufficiently address your concerns and lead to an improved rating of our work. Should you have any further questions or require additional clarification, please do not hesitate to ask.
>
> We look forward to your updated review and remain open to further discussions. Thanks so much.

---

> > ### Comment · Reviewer_TMW5 · 2024-11-28
> > **Authors Response**
> >
> > I thank the authors for their prompt and comprehensive response. I consider that my concerns have been addressed. I am happy to update my score.

---

> > > ### Author Response · Authors · 2024-11-29
> > > **Thank you**
> > >
> > > Thank you for reading our responses and for appreciating this work! Thanks again!

---

### Official Review · Reviewer_rMit · 2024-11-03

**Soundness:** 3
**Presentation:** 2
**Contribution:** 3
**Rating:** 5
**Confidence:** 4

**Summary:**

This work investigates whether large language models (LLMs) can autonomously generate novel and valid hypotheses in chemistry based solely on a research background. Specifically, the study explores if LLMs, when provided with a research question or background survey, can independently identify inspirations, synthesize hypotheses, and evaluate their quality. Building on discussions with chemistry experts, the authors hypothesize that most chemistry hypotheses can be derived from combining research background with relevant inspirations. They break down the central question into three key tasks: retrieving valuable inspirations, generating hypotheses, and ranking the quality of these hypotheses. To test this approach, the authors create a benchmark of 51 high-impact chemistry papers published in 2024. LLMs, trained only on data up to 2023, attempt to rediscover these hypotheses from the background and an extensive chemistry literature corpus. The results, achieved through a multi-agent LLM framework, show promising success, with many rediscovered hypotheses closely matching the originals and capturing core innovations.

**Strengths:**

1.	Generating research hypotheses is a complicated task, and the authors heuristically decomposed hypothesis generation into two steps:

(1). inspiration retrieval, and

(2). hypothesis refinement. In the hypothesis refinement step, the authors propose a novel “mutate and recombine” trick to help generate good hypotheses.

2.	The experiments to verify each of the research questions are well-designed with good quality.

**Weaknesses:**

1.	The introduction section could be written better and more clear. (a) It would be great if the authors could provide a summary of the major contributions of this work at the end of the introduction section. What are really the contribution to the field? (b) It would be great if the authors could briefly discuss why the decomposition of the major question is necessary, what’s the difference or connection between the proposed inspiration identification (the first step of the three) and Retrieval Augmented Generation (RAG).

2.	As the end goal is to rediscover the chemistry scientific hypotheses, the upper bound of “rediscovery” is the exact match of the original hypothesis, intuitively if we rank the original hypothesis with the generated hypothesis, the original hypothesis may be ranked at the top for most of the time. Experiment on this and analysis about the cases where the original hypothesis is not ranked at the top would be interesting to better reflect LLM’s ability to perform R(h).

3. Although the proposed approach constructed a benchmark as the basis to evaluate the performance, it is unclear to what degree the generated hypotheses can be trusted or reliable, not made up from baseless hallucination.

**Questions:**

See the weaknesses listed above.

---

> ### Author Response · Authors · 2024-11-20
> **Reply to Reviewer rMit - Part 1/2**
>
> Thank you for your insightful inquiries. In the following sections, we've structured our responses to each of your points raised.
>
>
> **Q1**: It would be great if the authors could provide a summary of the major contributions of this work
>
> **A1**: Sure! Overall, our contributions are:
> We provide the first mathematical derivation on how to decompose the seemingly-impossible-to-solve question P(hypothesis | background) into many executable and practical smaller steps This decomposition make P(hypothesis | background) possible to be practical;
> We develop a scientific discovery framework directly based on the mathematical derivation. Different from previous work, we propose an evolutionary algorithm based method to better associate background and inspiration, multi-step inspiration retrieval and composition, and an efficient ranking method. The framework can be applied to chemistry and material science, where previous methods only covered social science, NLP, and biomedical domains;
> We construct a benchmark by three chemistry PhD students, consisting of 51 chemistry papers published on Nature, Science, or a similar level, decomposing each paper into the research background, inspirations, and hypothesis.
> For the first time, we show that an LLM-based framework can rediscover chemistry hypotheses that deserve publication in Nature and Science. It is guaranteed that the rediscovery is not because of data contamination, because we have controlled the date of the training corpus of the LLM and the date of the chemistry papers that are available online.
> We will add them to the end of the introduction section.
>
>
> **Q2**: It would be great if the authors could briefly discuss why the decomposition of the major question is necessary.
>
>
> **A2**: Without the decomposition, we would need to solve the very challenging problem P(hypothesis | background) directly, which is too difficult to lead to good discovery.
>
> For example in this table, Qi’s method is to directly work on P(hypothesis | background), but the performance is very bad.
>
> Table 10: (Gemini-1.5-Pro)
>
> | Method                                          | Top Matched Score (by Gemini-1.5-Pro) | Average Matched Score (by Gemini-1.5-Pro) |
> |-------------------------------------------------|---------------------------------------|-------------------------------------------|
> | MOOSE                                           | 3.039                                 | 2.690                                     |
> | SciMON                  | 2.980                                 | 2.618                                     |
> | Qi                      | 2.216                                 | 1.846                                     |
> | MOOSE-Chem                                      | 3.686                                 | 2.443                                     |
> | w/o multi-step                                  | 3.588                                 | 2.529                                     |
> | w/o multi-step and w/o mutation & recombination | 2.902                                 | 2.631                                     |
>
> By decomposing the very challenging problem into many practical smaller steps, we can solve the very challenging problem by solving each small step. We have shown it’s effectiveness through experiments.
>
>
> **Q3**: What’s the difference or connection between the proposed inspiration identification (the first step of the three) and Retrieval Augmented Generation (RAG).
>
> **A3**: They are all retrieval methods, but with different goals. RAG would most likely retrieve those information that is semantically the most similar to the input information (research background), while here our goal is to retrieve those information that was not known to be related to the input information before, but in fact is related. We assume that LLMs might have the ability to do it.

---

> ### Author Response · Authors · 2024-11-20
> **Reply to Reviewer rMit - Part 2/2**
>
> **Q4**: Experiment on the ranking of the groundtruth hypothesis and analysis about the cases where the original hypothesis is not ranked at the top would be interesting to better reflect LLM’s ability to perform R(h)
>
> **A4**: Thank you for the suggestion. We have conducted experiments to understand the ranking ratio of the ground truth hypothesis. The results are shown below:
>
> |                    | Overall | Validness | Novelty | Significance | Potential |
> |--------------------|---------|-----------|---------|--------------|-----------|
> | Average Rank Ratio | 0.65    | 0.75      | 0.76    | 0.73         | 0.70      |
>
> The rank ratio is the lower the better. This result indicates that LLM does poorly on ranking hypotheses. It is consistent with the observations in Table 9, where those hypotheses receiving a high matched score do not get an apparent edge than other hypotheses. One of the reasons that the ground truth hypothesis is lower than average could be that the generated hypotheses might describe its novelty and significance (although they are prompted to not to), which might influence the judgment.
>
> **Q5**: Although the proposed approach constructed a benchmark as the basis to evaluate the performance, it is unclear to what degree the generated hypotheses can be trusted or reliable, not made up from baseless hallucination.
>
> **A5**: Here we (1) have chemistry PhD students for the evaluation; (2) adopt reference-based evaluation (groundtruth hypothesis is provided, so that PhD students only need to compare the generated hypothesis and the groundtruth hypothesis, which should be easier). In fact, we count on LLM’s hallucination ability to find novel hypotheses: a novel hypothesis would not have been observed by the LLMs, therefore all novel hypotheses come from the class of hallucination.
>
>
> **We greatly appreciate your comprehensive feedback. We hope that our responses have satisfactorily addressed all your queries. Should you have further questions or suggestions for enhancing our manuscript, we warmly welcome your input.**

---

> ### Author Response · Authors · 2024-11-22
> **Looking forward to your reply**
>
> Dear Reviewer,
>
> Thank you for your insightful comments and suggestions. We have submitted responses to address the points you've raised, including clarifications and additional experimental results. We hope that these will sufficiently address your concerns and lead to an improved rating of our work. Should you have any further questions or require additional clarification, please do not hesitate to ask.
>
> We look forward to your updated review and remain open to further discussions. Thanks so much.

---

> > ### Comment · Reviewer_rMit · 2024-11-25
> >
> > We thank the authors for their reply and appreciate the additional experiments conducted. We encourage the authors to update the paper promptly to reflect the discussions with the reviewers. For instance, the extra experiments and discussions could be included in the appendix, while the key contributions could be summarized in the introduction.
> >
> > We read other reviewer's concerns, and they raised several insightful points. As Reviewer zDBz mentioned, using only 51 publications might not provide a statistically significant foundation for a comprehensive benchmark to compare models and techniques. Expanding the benchmark to include other chemistry papers—such as those with notable discoveries or high citation counts published after 2024—could enhance its robustness.
> >
> > Regarding the ranking of hypotheses by LLMs, SCIMUSE demonstrates that LLM rankings align well with human expert opinions. If an original hypothesis ranks lower than the generated ones, this suggests that the generated hypotheses may surpass the original in quality. This finding may help to shift the focus from "rediscovery" to generating novel, high-quality hypotheses, which is more practical, forward-looking, and beneficial for scientific discovery. In contrast, rediscovery might appear more retrospective, serving primarily as a proof-of-concept without clear practical applications for advancing science.
> >
> > [SCIMUSE]Gu, X., & Krenn, M. (2024). Generation and human-expert evaluation of interesting research ideas using knowledge graphs and large language models. arXiv preprint arXiv:2405.17044.

---

> ### Author Response · Authors · 2024-11-25
> **Reply to Reviewer rMit**
>
> Thank you for carefully reading our responses and appreciating our reply and the additional experiments conducted!
> As you suggested, we have added the extra experiments and discussions in the appendix, and summarized the key contributions in the introduction.
>
> **Q1**: 51 publications might not provide a statistically significant foundation for a comprehensive benchmark to compare models and techniques
> **A1**: Firstly, in the scientific discovery tasks, benchmarks involving several dozen high-quality publications are considered sufficient for meaningful analysis. For instance, previous works like [1] which analyzed 50 publications, [2] with 44, and [3] with 20, have set precedents for what can be achieved with focused, quality-driven datasets. Our selection includes 51 papers from prestigious journals like Nature and Science, which we believe should be considered as notable discoveries.
>
> [1] Large language models for automated open-domain scientific hypotheses discovery, ACL 2024
> [2] ScienceAgentBench: Toward Rigorous Assessment of Language Agents for Data-Driven Scientific Discovery
> [3] DiscoveryBench: Towards Data-Driven Discovery with Large Language Models
>
> Secondly, the primary goal of our benchmark was not to achieve comprehensive coverage but to demonstrate a proof of concept for our system's capability to rediscover significant chemical hypotheses (corresponds to the fourth contribution).
>
> Thirdly, the benchmark drew parallels with the Mathematical Olympiad Competition to illustrate our methodology. Here, the focus is on solving a set number of highly challenging problems. Just as solving 25 out of 30 complex math problems in an Olympiad can earn a gold medal [4], rediscovering 40 significant hypotheses published in Nature and Science out of 51 could be seen as an equally or more impactful achievement due to its real-world implications.
>
> [4] Trinh, T.H., Wu, Y., Le, Q.V. et al. Solving olympiad geometry without human demonstrations. Nature 2024
>
> **Q2**: SCIMUSE demonstrates that LLM rankings align well with human expert opinions. If an original hypothesis ranks lower than the generated ones, this suggests that the generated hypotheses may surpass the original in quality.
>
> **A2**: Yes, we agree with the possibility. We have cited SCIMUSE and discuss the possibilities in Appendix A.7, line 996~1005.
>
> **Q3**: This finding may help to shift the focus from "rediscovery" to generating novel, high-quality hypotheses, which is more practical, forward-looking, and beneficial for scientific discovery. In contrast, rediscovery might appear more retrospective, serving primarily as a proof-of-concept without clear practical applications for advancing science.
>
> **A3**: The setting of this paper is to "generating novel, high-quality hypotheses". We are glad that you appreciate this setting. The proposed framework is designed to, and can already be used to generate novel, high-quality hypotheses. In the simplest case, the framework only need to take a chemistry research question (in any domain of chemistry), and the framework can generate a list of hypotheses (ranked). Supported by the experiment observations, it is very likely that the list will include the novel and high-quality hypotheses.
>
> We work on rediscovery mainly for the evaluation purpose: otherwise there would be no strongly persuading ways for evaluation, except for conducting chemistry experiments.
>
> In addition, this research field is still in a young stage, while no previous works have even formally worked on rediscovery.
> Please refer to our fourth contribution: "For the first time, we show that an LLM-based framework can largely rediscover many chemistry hypotheses that have published in Nature and Science. It is guaranteed that the rediscovery is not because of data contamination, because we have controlled the date of the training corpus of the LLM and the date of the chemistry papers that are available online."
> There has been neither no works that have ever included a chemistry experiment conduction for verification, because of the lack of available chemistry resources.
>
> But we do agree that collaboration with chemistry team should be a future direction. In fact we are reaching out to chemistry teams for collaboration, but it takes time.
>
> **We hope the above replies are clear enough to address the reviewer's concern.** If the reviewer still has concerns about the rating of our work (which is 5 right now), we are glad to have further discussion. Should you find our clarifications satisfactory, we kindly hope for a reconsideration of the rating for our paper. Thanks again for your detailed review and valuable feedback.

---

> > ### Author Response · Authors · 2024-11-28
> > **Hoping to continue the discussion**
> >
> > Dear Reviewer,
> >
> > We appreciate your participation in this productive discussion. We hope all your doubts have been addressed. If so, we humbly request that you consider increasing the score.
> >
> > We also encourage you to review our responses to other reviewers who raised important points and chose to increase their scores after being satisfied with our explanations and additional experiments.
> >
> > Thank you!
> >
> > --
> >
> > Authors

---

> > > ### Author Response · Authors · 2024-12-02
> > > **Requesting to check our last response**
> > >
> > > Dear Reviewer,
> > >
> > > We appreciate you finding time to engage with us despite your busy schedule.
> > > Since today is the final day for submitting rebuttals, we hope you had the opportunity to read our last response. We have made every effort to address all our concerns. Therefore, we kindly ask if you could provide any last thoughts or, if you feel it's warranted based on our revisions and clarifications, consider updating your rating score.
> > >
> > > Your final input would be invaluable in ensuring a comprehensive review process. Thank you again for your time and expertise throughout this discussion.

---

### Official Review · Reviewer_zDBz · 2024-11-03

**Soundness:** 3
**Presentation:** 3
**Contribution:** 2
**Rating:** 6
**Confidence:** 3

**Summary:**

This paper investigates the potential of LLMs to automatically discover novel and valid hypotheses in the field of chemistry. The study introduces a framework called MOOSE-Chem, which decompose the main research question into three smaller questions, respectively focusing on (1) the identification of inspiration papers; (2) the inference of unknown knowledge that is highly likely to be valid; and (3) the identification and ranking of hypotheses generated by the LLMs.

**Strengths:**

Originality:
Firstly, while LLMs have been utilized for scientific discovery in social science and NLP, this paper is the first to investigate their potential in chemistry.
Besides, The MOOSE-CHEM framework employs a three-step approach to retrieve inspiration papers, inference valid knowledge, identify hypotheses and rank them, which hasn’t been used in previous research.
Moreover, the use of the evolutionary algorithm to foster a broader diversity in hypothesis generation is also an innovation point.

Quality:
The paper includes extensive comparative experiments and ablation studies, providing a thorough evaluation of LLMs' performance. It also demonstrates the consistency between expert evaluation and automated evaluation of MS, enhancing the reliability of the results.

Clarity:
The methodology is well-structured, and the three steps based on smaller questions are well-defined. Besides, detailed evaluation methods and corresponding formulas are provided for the three research questions, Q1, Q2 and Q3.  The practical implementation process is also relatively clear.

Significance:
The ability of LLMs to generate hypotheses can significantly accelerate the pace of scientific discovery by reducing the time and effort required for hypothesis generation. It can also open researchers’ minds for the application of LLMs in scientific discovery.

**Weaknesses:**

Firstly, using the same large language model to evaluate its own generated results may introduce bias. It is recommended to try using different LLMs to evaluate the results so as to guarantee the reliability of the results. For example, consider using models like LLaMa[1], Claude[2], Gemini[3], or other recent LLMs to compare outputs. If using the same LLM is necessary, you could collect hypotheses generated by humans and also have both experts and GPT-4 evaluate them. Then, compare their Hard/Soft Consistency Scores as well as distribution of MS with those of the hypotheses generated by the LLM.

[1] Touvron H, Martin L, Stone K, et al. Llama 2: Open foundation and fine-tuned chat models[J]. arXiv preprint arXiv:2307.09288, 2023.
[2] Anthropic. The claude 3 model family: Opus, sonnet, haiku. https://www.anthropic.com, 2024.
[3] Team G, Anil R, Borgeaud S, et al. Gemini: a family of highly capable multimodal models[J]. arXiv preprint arXiv:2312.11805, 2023.

Besides, during the evaluation process (as demonstrated in section 5.1), in the constructed dataset, other papers besides the ground truth inspiration papers may also be helpful for the current background. This step may also require further evaluation to obtain more reliable results. You might select several high-ranking papers that are not included in set I, particularly those that repeatedly appear in experiments with varying corpus sizes or screen window sizes, and conduct case studies on them.

Additionally, this paper only employs the titles and abstracts for retrieval, which might overlook inspirations that could arise from the detailed content of the articles. Sections like Conclusion or Discussion often contain insights into the paper's limitations and suggestions for future work. These elements are usually crucial for generating inspiration.

Moreover, the benchmarks mentioned in the paper contain only 51 references, which may not be sufficient to fully assess the effectiveness of the proposed method. You are advised to increase the number of samples, especially selecting literature from different chemical fields. In addition to traditional categories such as Analytical Chemistry, Organic Chemistry, and Inorganic Chemistry, broader research topics like Environmental Chemistry, Medicinal Chemistry, and Nuclear Chemistry can also be considered. Furthermore, Nature Chemistry includes interdisciplinary areas such as Bioinorganic Chemistry, Bioorganic Chemistry, and Organometallic Chemistry, which could be investigated as well. Of course, this may involve considerations related to the resource consumption of LLMs.

Finally, although the paper used chemistry PhD students for the evaluation, there is still subjectivity that may affect the reliability of the evaluation results. You might consider inviting more experts to participate in the evaluation and analyzing the consistency of the assessments from different human individuals.

**Questions:**

1. There is a lack of experimental exploration for mutation settings.   I am curious about the impact of mutations on the final results.
     For example, according to the ablation studies, what proportion of high-quality hypotheses can be obtained directly without mutations?

2. How can we explain that a smaller window size leads to better performance in the inspiration retrieval phase?

3. Did the authors replace GPT-4o for all comparison methods for generation? Please specify.

4. How helpful is it to add new dimensions of significance?

---

> ### Author Response · Authors · 2024-11-20
> **Reply to Reviewer zDBz - Part 1/3**
>
> We appreciate your thoughtful questions and suggestions. Below, we have organized your queries along with our responses for clarity:
>
> **Q1**: It is recommended to try using different LLMs in addition to GPT-4o to evaluate the results so as to guarantee the reliability of the results.
>
> **A1**: Thank you for the advice. We have additionally used Claude-3.5-Sonnet and Gemini-1.5-pro for all of the evaluation in this paper. Specifically, Table 5 and Table 10 used GPT-4o as an evaluation method. Here we show Table 5 and Table 10 in terms of Claude-3.5-Sonnet based and Gemini-1.5-pro based evaluation.
> The results show that the previous conclusions are still valid and supported.
>
>
> Table 5: (Claude-3.5-Sonnet)
>
> |                                | 5  | 4  | 3  | 2  | 1 | 0 |
> |--------------------------------|----|----|----|----|---|---|
> |      w/ background survey      |    |    |    |    |   |   |
> | Top MS (Claude-3.5-Sonnet)     | 33 | 7  | 10 | 1  | 0 | 0 |
> | Average MS (Claude-3.5-Sonnet) | 4  | 19 | 15 | 10 | 3 | 0 |
> |      w/o background survey     |    |    |    |    |   |   |
> | Top MS (Claude-3.5-Sonnet)     | 31 | 19 | 1  | 0  | 0 | 0 |
> | Average MS (Claude-3.5-Sonnet) | 7  | 24 | 18 | 2  | 0 | 0 |
>
>
> Table 5: (Gemini-1.5-Pro)
>
> |                             | 5  | 4  | 3  | 2  | 1  | 0 |
> |-----------------------------|----|----|----|----|----|---|
> |     w/ background survey    |    |    |    |    |    |   |
> | Top MS (Gemini-1.5-Pro)     | 20 | 18 | 0  | 12 | 1  | 0 |
> | Average MS (Gemini-1.5-Pro) | 2  | 13 | 17 | 8  | 11 | 0 |
> |    w/o background survey    |    |    |    |    |    |   |
> | Top MS (Gemini-1.5-Pro)     | 19 | 19 | 1  | 11 | 0  | 1 |
> | Average MS (Gemini-1.5-Pro) | 4  | 9  | 14 | 15 | 5  | 4 |
>
>
>
>
>
> Table 10: (Claude-3.5-Sonnet)
> | Method                                          | Top Matched Score (by Claude-3.5-Sonnet ) | Average Matched Score (by Claude-3.5-Sonnet ) |
> |-------------------------------------------------|-------------------------------------------|-----------------------------------------------|
> | MOOSE                                           | 3.902                                     | 3.559                                         |
> | SciMON                  | 3.824                                     | 3.529                                         |
> | Qi                      | 3.431                                     | 3.092                                         |
> | MOOSE-Chem                                      | 4.471                                     | 3.697                                         |
> | w/o multi-step                                  | 4.216                                     | 3.592                                         |
> | w/o multi-step and w/o mutation & recombination | 3.941                                     | 3.614                                         |
>
>
> Table 10: (Gemini-1.5-Pro)
>
> | Method                                          | Top Matched Score (by Gemini-1.5-Pro) | Average Matched Score (by Gemini-1.5-Pro) |
> |-------------------------------------------------|---------------------------------------|-------------------------------------------|
> | MOOSE                                           | 3.039                                 | 2.690                                     |
> | SciMON                  | 2.980                                 | 2.618                                     |
> | Qi                      | 2.216                                 | 1.846                                     |
> | MOOSE-Chem                                      | 3.686                                 | 2.443                                     |
> | w/o multi-step                                  | 3.588                                 | 2.529                                     |
> | w/o multi-step and w/o mutation & recombination | 2.902                                 | 2.631                                     |
>
>
>
>
> **Q2**: Other papers besides the ground truth inspiration papers may also be helpful for the current background.
>
>
> **A2**: Yes, we agree. The existing dataset has been constructed to include those not ground truth inspiration papers. Specifically, we found non-groundtruth inspiration papers that discuss a similar topic with the groundtruth inspiration papers. The existing benchmark has included many of these non-groundtruth inspiration papers. To avoid ambiguity, the “Reasoning Process” tab in the benchmark indicates the relation between the collected inspirations and hypothesis, e.g., “hypothesis = inspiration 1/2 + inspiration 3”. While there’s a repetition (“1/2”), it usually indicates one non-groundtruth inspiration paper we found. In general, more than 20% of the inspirations are non-groundtruth inspirations.

---

> ### Author Response · Authors · 2024-11-20
> **Reply to Reviewer zDBz - Part 2/3**
>
> **Q3**: This paper only employs the titles and abstracts for retrieval, which might overlook inspirations that could arise from the detailed content of the articles.
>
> **A3**: The objective of our paper is to demonstrate that Large Language Models (LLMs), trained on data up to December 2023, possess the capability to rediscover hypotheses that were published in Nature in 2024, which were exclusively available online during that year. Utilizing merely the titles and abstracts, we have effectively validated our hypothesis. While we concede that incorporating comprehensive details from the source papers could be beneficial, such depth is unnecessary for addressing the core research question we explore. We invite the research community to delve deeper into this promising avenue of investigation.
>
>
>
>
> **Q4**: The benchmarks mentioned in the paper contain only 51 publications.
>
> **A4**: The benchmark is designed to be similar to the Mathematical Olympiad Competition, to provide several dozens of very difficult and meaningful questions to solve.
> We consider that publishing on Nature is at least on the same intellectual level as solving an Olympiad question, but being more meaningful to the world.
> For the math Olympiad contest, only 30 questions need to be answered [1], and here we have 51.
>
> [1] Trinh, T.H., Wu, Y., Le, Q.V. et al. Solving olympiad geometry without human demonstrations. Nature 625, 476–482 (2024). https://doi.org/10.1038/s41586-023-06747-5
>
>
> **Q5**: In addition to traditional categories such as Analytical Chemistry, Organic Chemistry, and Inorganic Chemistry, broader research topics like Environmental Chemistry, Medicinal Chemistry, and Nuclear Chemistry can also be considered. Furthermore, Nature Chemistry includes interdisciplinary areas such as Bioinorganic Chemistry, Bioorganic Chemistry, and Organometallic Chemistry, which could be investigated as well.
>
> **A5**: Respected reviewer, you are absolutely correct, and in fact, your perspective aligns closely with what we have done. In the field of chemistry, it is generally acknowledged that there are four major branches: organic chemistry, analytical chemistry, inorganic chemistry, and physical chemistry. We categorize the existing 51 papers in the benchmark according to the major branches. With the advancement of the discipline, numerous interdisciplinary subfields have emerged, such as biochemistry and organometallic chemistry.
>
> In the literature we selected, very few studies adopt a purely single-disciplinary approach. For example, while we have broadly categorized some of the benchmark papers as "organic chemistry," the works we reviewed span several specialized areas. For instance, some belong to organometallic chemistry, others to electrocatalysis, computational chemistry, or photocatalysis. In fact, some studies fall under both photocatalysis and organometallic chemistry. In addition to the papers we categorized as “organic chemistry”, other papers can fall in the domain of Environmental Chemistry, Nuclear Chemistry, Material Chemistry, and Energy Chemistry. Therefore, the benchmark is versatile and has covered many of these chemistry subfields.

---

> ### Author Response · Authors · 2024-11-20
> **Reply to Reviewer zDBz - Part 3/3**
>
> **Q6**: Although the paper used chemistry PhD students for the evaluation, there is still subjectivity that may affect the reliability of the evaluation results.
>
> **A6**: Yes, we agree. Here we (1) have chemistry PhD students for the evaluation; (2) adopt reference-based evaluation (groundtruth hypothesis is provided, so that PhD students only need to compare the generated hypothesis and the groundtruth hypothesis, which should be easier). We have invited three Chemistry PhD students for the evaluation, and we find that the expert consistency is as below:
>
> | Hard Consistency Score  | Soft Consistency Score  |
> |-------------------------|-------------------------|
> | 0.458                   | 0.958                   |
>
> Hard consistency is assigned to 1 only if the two scores are exactly the same, else is assigned to 0. Soft consistency is assigned to 1 only if the absolute difference between the two scores is less than 2, else is assigned to 0.
> The consistency score shows that experts in general share a high-level consistency, meaning that the evaluation results are highly reliable.
>
>
>
> **Q7**: According to the ablation studies, what proportion of high-quality hypotheses can be obtained directly without mutations?
>
> **A7**: Thank you for noticing the ablation study on mutations in Table 10, which shows that mutations can help to improve the performance on discovery.
>
> | Matched Score Threshold | only non EA branch | only EA branches
> |-----------|--------------------|-----------------|
> |         5 |                 16 |              46 |
> |         4 |                 19 |              54 |
>
>
> This table shows the number of high-quality hypotheses in terms of the non evolutionary algorithm (EA) branch, only EA branches, and only the final recombination EA branch. It indicates that about one third of high quality hypotheses can be obtained directly without mutations.
>
>
> **Q8**: How can we explain that a smaller window size leads to better performance in the inspiration retrieval phase?
>
> **A8**: Thank you for noticing the ablation study on window size in Table 4, which is to understand the relation of window size to the retrieval performance. The reason is probably because larger window size means more contents in the input, while LLMs are not good at processing a (much) longer input compared to the shorter input.
>
> **Q9**: Did the authors replace GPT-4o for all comparison methods for generation? Please specify.
>
> **A9**: Yes, please refer to the A1 answer.
>
> **Q10**: How helpful is it to add new dimensions of significance?
>
> **A10**: Thank you for the question. We have conducted an experiment without significance feedback, and we find that our method even performs better in terms of the matched score.
> We attribute this phenomenon to the LLM’s creativity: when asked to generate significant hypotheses, LLMs tend to largely deviate from the existing information for more possible significance, resulting in a lower matched score. However, the matched score only measures the match degree of one given groundtruth hypothesis, and it is possible that the more deviated one is more significant.
>
> |                                | 5  | 4  | 3  | 2  | 1 | 0 |
> |--------------------------------|----|----|----|----|---|---|
> |    w/ significance feedback    |    |    |    |    |   |   |
> | Top MS (Claude-3.5-Sonnet)     | 33 | 7  | 10 | 1  | 0 | 0 |
> | Average MS (Claude-3.5-Sonnet) | 4  | 19 | 15 | 10 | 3 | 0 |
> |    w/o significance feedback   |    |    |    |    |   |   |
> | Top MS (Claude-3.5-Sonnet)     | 34 | 13 | 4  | 0  | 0 | 0 |
> | Average MS (Claude-3.5-Sonnet) | 8  | 28 | 11 | 3  | 1 | 0 |
>
>
>
> **We greatly appreciate your comprehensive feedback. We hope that our responses have satisfactorily addressed all your queries. Should you have further questions or suggestions for enhancing our manuscript, we warmly welcome your input.**

---

> ### Author Response · Authors · 2024-11-22
> **Looking forward to your reply**
>
> Dear Reviewer,
>
> Thank you for your insightful comments and suggestions. We have submitted responses to address the points you've raised, including clarifications and additional experimental results. We hope that these will sufficiently address your concerns and lead to an improved rating of our work. Should you have any further questions or require additional clarification, please do not hesitate to ask.
>
> We look forward to your updated review and remain open to further discussions. Thanks so much.

---

> > ### Author Response · Authors · 2024-11-28
> > **Our humble request to read our rebuttal**
> >
> > Dear Reviewer,
> >
> > We appreciate your productive comments. We hope all your doubts have been addressed. If so, we humbly request that you consider increasing the score.
> >
> > We also encourage you to review our responses to other reviewers who raised important points and chose to increase their scores after being satisfied with our explanations and additional experiments.
> >
> > Thank you!
> >
> > --
> >
> > Authors

---

> ### Author Response · Authors · 2024-12-02
> **Requesting to read our response**
>
> Dear Reviewer,
>
> Since today is the final day for submitting rebuttals, we hope you had the opportunity to read our responses. We have made every effort to address all the concerns. If you find that we have resolved them, we would appreciate it if you could consider increasing the score. Thank you!

---

> > ### Comment · Reviewer_zDBz · 2024-12-02
> >
> > Thanks for the detailed responses to address my concerns. I have raised the score.

---

> > > ### Author Response · Authors · 2024-12-02
> > >
> > > Thank you for reading our responses! Glad to know that your concerns have been addressed!

---

### Author Response · Authors · 2024-11-21
**Thank All Reviewers**

To all reviewers:

We would like to thank all reviewers for their thoughtful insights and valuable comments.

It seems that the vacancy of a summarization of the contributions of this paper have caused many misunderstandings. We summarize here and will add them to the paper:

1. We provide the first mathematical derivation on how to decompose the seemingly-impossible-to-solve question P(hypothesis | research background) into many executable and practical smaller steps. This decomposition make P(hypothesis | research background) possible to be practical;
2. We develop a scientific discovery framework directly based on the mathematical derivation. Different from previous work, we propose an evolutionary algorithm based method to better associate background and inspiration, multi-step inspiration retrieval and composition, and an efficient ranking method. In addition, the framework can be applied to chemistry and material science, where previous methods only covered social science, NLP, and biomedical domains;
3. We construct a benchmark by three chemistry PhD students, consisting of 51 chemistry papers published on Nature, Science, or a similar level, decomposing each paper into the research background, inspirations, and hypothesis;
4. **For the first time, we show that an LLM-based framework can largely rediscover many chemistry hypotheses that have published in Nature and Science**. It is guaranteed that the rediscovery is not because of data contamination, because we have controlled the date of the training corpus of the LLM and the date of the chemistry papers that are available online.

We are excited that you recognized our contributions. We quote correspondingly as below:

1. "the authors did a good job guiding the reader through the motivation, the derivation of the method and motivating each of the proposed steps and experiments" [reviewer TMW5]; "Generating research hypotheses is a complicated task, and the authors heuristically decomposed hypothesis generation into ..." [reviewer rMit]; "corresponding formulas are provided for the three research questions, Q1, Q2 and Q3." [reviewer zDBz]
2. "The MOOSE-CHEM framework employs a three-step approach to retrieve inspiration papers, inference valid knowledge, identify hypotheses and rank them, which hasn’t been used in previous research." [reviewer zDBz]; "The use of beam search and evolutionary algorithms for hypothesis refinement seems simple and powerful" [reviewer km2P]; "this paper is the first to investigate LLM's potential in chemistry" [reviewer zDBz];
3. "To test this approach, the authors create a benchmark of 51 high-impact chemistry papers published in 2024." [reviewer rMit]; "The authors gathered a carefully curated dataset of scientific papers where questions and hypotheses were extracted for evaluation."[reviewer TMW5]
4. "The results, ..., show promising success, with many rediscovered hypotheses closely matching the originals and capturing core innovations." [reviewer rMit]

We are grateful that you also found that:
1. The paper is generally well-written, and in good English. [reviewer TMW5]
2. The experiments to verify each of the research questions are well-designed with good quality. [reviewer rMit]
3. Every claim made on the performance of the method is generally backed up with experiments. [reviewer TMW5]
4. This paper provides a thorough evaluation of LLM's performance on the scientific discovery task. [reviewer zDBz]
5. Related work is well covered, and experiments are included that compare the proposed method with previous work. [reviewer TMW5]


We also appreciate many helpful suggestions, based on which we have improved our manuscript. The main changes are:
1. We added Claude-3.5-Sonnet and Gemini-1.5-pro for automatic evaluation for all experiments, replacing GPT-4o.
2. We added the consistency score between experts for expert evaluation.
3. We analyzed the proportion of high-quality hypotheses from the mutation branch and not from the mutation branch.
4. We added the analyse on the effect of the significance feedback.
5. We added the ranking ratio of ground truth hypotheses.
6. We added the results of random screening baseline for inspiration retrieval.
7. We added different size of llama's experiments (Llama-3.1-8b, Llama-3.1-70b, and Llama-3.1-405b), in addition to GPT-4o.


We would again like to thank all reviewers for their time and effort, and we hope that our changes adequately address all concerns. We are glad to have further discussions and take additional suggestions to help improve our manuscript.

Sincerely,

Authors

---

### Meta-Review · Area_Chair_it6Q · 2024-12-12

**Metareview:**

The authors propose to combine LLMs with traditional search strategies for generating/refining hypotheses in scientific domains (e.g., chemistry). Hence, the problem setting is very challenging and rather unexplored until now. The technical solution may not be the end of the story but appropriate. The paper has also been well received by the reviewers. Particularly the empirical evaluation is strong and provides convincing results.

**Additional Comments On Reviewer Discussion:**

Not every reviewer got involved in discussion with the authors but those who did were satisfied with the rebuttal.

---

### Decision · Program_Chairs · 2025-01-22

Accept (Poster)